# Optimal Epoch Stochastic Gradient Descent Ascent Methods for Min-Max Optimization

**Yan Yan**
School of EECS
Washington State University
yanyan.1@wsu.edu

**Yi Xu**
Machine Intelligence Technology
Alibaba Group US Inc
statxy@gmail.com

**Qihang Lin**
Department of Business Analytics
University of Iowa
qihang-lin@uiowa.edu

**Wei Liu**
Tencent AI Lab
wl2223@columbia.edu

**Tianbao Yang**
Department of CS
University of Iowa
tianbao-yang@uiowa.edu

## Abstract

Epoch gradient descent method (a.k.a. Epoch-GD) proposed by [16] was deemed a breakthrough for stochastic strongly convex minimization, which achieves the optimal convergence rate of $O(1/T)$ with $T$ iterative updates for the *objective gap*. However, its extension to solving stochastic min-max problems with strong convexity and strong concavity still remains open, and it is still unclear whether a fast rate of $O(1/T)$ for the *duality gap* is achievable for stochastic min-max optimization under strong convexity and strong concavity. Although some recent studies have proposed stochastic algorithms with fast convergence rates for min-max problems, they require additional assumptions about the problem, e.g., smoothness, bi-linear structure, etc. In this paper, we bridge this gap by providing a sharp analysis of epoch-wise stochastic gradient descent ascent method (referred to as Epoch-GDA) for solving strongly convex strongly concave (SCSC) min-max problems, without imposing any additional assumption about smoothness or the function's structure. To the best of our knowledge, our result is the first one that shows Epoch-GDA can achieve the optimal rate of $O(1/T)$ for the duality gap of general SCSC min-max problems. We emphasize that such generalization of Epoch-GD for strongly convex minimization problems to Epoch-GDA for SCSC min-max problems is non-trivial and requires novel technical analysis. Moreover, we notice that the key lemma can also be used for proving the convergence of Epoch-GDA for weakly-convex strongly-concave min-max problems, leading to a nearly optimal complexity without resorting to smoothness or other structural conditions.

## 1 Introduction

In this paper, we consider stochastic algorithms for solving the following min-max saddle-point problem with a general objective function $f$ *without smoothness or any other special structure*:

$$\min_{x \in X} \max_{y \in Y} f(x, y), \tag{1}$$

where $X \subseteq \mathbb{R}^d$ and $Y \subseteq \mathbb{R}^n$ are closed convex sets and $f : X \times Y \to \mathbb{R}$ is continuous. It is of great interest to find a saddle-point solution to the above problem, which is defined as $(x^*, y^*)$ such that $f(x^*, y) \leq f(x^*, y^*) \leq f(x, y^*), \forall x \in X, y \in Y$. Problem (1) covers a number of applications

in machine learning, including distributionally robust optimization (DRO) [31, 30], learning with non-decomposable loss functions [27, 11, 43, 26], and generative adversarial networks [13, 3].

In this work, we focus on two classes of the min-max problems: (i) strongly-convex strongly-concave (SCSC) problem where $f$ is strongly convex in terms of $x$ for any $y \in Y$ and is strongly concave in terms of $y$ for any $x \in X$; (ii) weakly-convex strongly-concave (WCSC) problem, where there exists $\rho > 0$ such that $f(x,y) + \frac{\rho}{2}\|x\|^2$ is strongly convex in terms of $x$ for any $y \in Y$ and is strongly concave in terms of $y$ for any $x \in X$. Both classes have applications in machine learning [41, 36].

Although stochastic algorithms for convex-concave min-max problems have been studied extensively in the literature, their research is still far behind its counterpart for stochastic convex minimization problems. Below, we highlight some of these gaps to motivate the present work. For the sake of presentation, we first introduce some terminologies. The duality gap at $(x, y)$ is defined as $\text{Gap}(x, y) := f(x, \hat{y}(x)) - f(\hat{x}(y), y)$, where $\hat{x}(y) := \arg\min_{x' \in X} f(x', y)$ and $\hat{y}(x) := \arg\max_{y' \in Y} f(x, y')$. If we denote by $P(x) := \max_{y' \in Y} f(x, y')$, then $P(x) - P(x^*)$ is the primal objective gap, where $x^* = \arg\min_{x \in X} P(x)$.

When $f$ is convex in $x$ and concave in $y$, many studies have designed and analyzed stochastic primal-dual algorithms for solving the min-max problems under different conditions of the problem (see references in next section). A standard result is provided by [32], which proves that primal-dual SGD suffers from a convergence rate of $O(1/\sqrt{T})$ for the duality gap without imposing any additional assumptions about the objective function. This is analogous to that for stochastic convex minimization [32]. However, the research of stochastic algorithms for SCSC problems lacks behind that for strongly convex minimization problems. A well-known result for stochastic strongly convex minimization is given by [16], which presents the first fast convergence rate $O(1/T)$ for stochastic strongly convex minimization by the Epoch-GD algorithm, which runs standard SGD in an epoch-wise manner by decreasing the step size geometrically. However, a fast rate of $O(1/T)$ for the *duality gap* of a stochastic algorithm **is still unknown for general SCSC problems**. We notice that there are extensive studies about stochastic algorithms with faster convergence rates than $O(1/\sqrt{T})$ for solving convex-concave min-max problems [46, 38, 37, 10, 6, 5, 35, 21, 41, 18, 47]. However, these works usually require additional assumptions about the objective function (e.g., smoothness, bilinear structure) or only prove the convergence in weaker measures (e.g., the primal objective gap, the distance of a solution to the saddle point).

We aim to bridge this gap by presenting the first optimal rate $O(1/T)$ of the duality gap for solving general SCSC problems. In particular, we propose an epoch-wise stochastic gradient descent ascent (Epoch-GDA) algorithm - a primal-dual variant of Epoch-GD that runs stochastic gradient descent update for the primal variable and stochastic gradient ascent update for the dual variable for solving (1). Although the algorithmic generalization is straightforward, the proof of convergence in terms of the duality gap for Epoch-GDA is not straightforward at all. We note that the key difference in the analysis of Epoch-GDA is that to upper bound the duality gap of a solution $(\bar{x}, \bar{y})$ we need to deal with the distance of an initial solution $(x_0, y_0)$ to the reference solutions $(\hat{x}(\bar{y}), \hat{y}(\bar{x}))$, where $\hat{x}(\bar{y}) = \arg\min_{x' \in X} f(x', \bar{y})$ and $\hat{y}(\bar{x}) = \arg\max_{y' \in Y} f(\bar{x}, y')$ depend on $\bar{y}$ and $\bar{x}$, respectively. In contrast, in the analysis of the objective gap for Epoch-GD, one only needs to deal with the distance from an initial solution $x_0$ to the optimal solution $x^*$, i.e., $\|x_0 - x^*\|_2^2$, which by strong convexity can easily connects to the objective gap $P(x_0) - P(x^*)$, leading to the telescoping sum on the objective gap. Towards addressing the challenge caused by dealing with the duality gap, we present a key lemma that connects the distance measure $\|x_0 - \hat{x}(\bar{y})\|_2^2 + \|y_0 - \hat{y}(\bar{x})\|_2^2$ to the duality gap of $(x_0, y_0)$ and $(\bar{x}, \bar{y})$. In addition, since we use the same technique as Epoch-GD for handling the variance of stochastic gradient by projecting onto a bounded ball with shrinking radius, we have to carefully prove that such restriction does not affect the duality gap for the original problem, which also needs to deal with bounding $\|x_0 - \hat{x}(\bar{y})\|_2^2$ and $\|y_0 - \hat{y}(\bar{x})\|_2^2$.

Moreover, we notice that the aforementioned key lemma and the telescoping technique based on the duality gap can also be used for proving the convergence of Epoch-GDA for **finding an approximate stationary solution of general WCSC problems**. The algorithmic framework is similar to that proposed by [36], i.e., by solving SCSC problems successively, but with a subtle difference in handling the dual variable. In particular, we do not need additional condition on the structure of the objective function and extra care for dealing with the dual variable for restart as done in [36]. This key difference is caused by our sharper analysis, i.e., we use the telescoping sum based on the duality gap instead of the primal objective gap as in [36]. As a result, our algorithm and analysis lead to a

Table 1: Summary of complexity results of this work and previous works for finding an $\epsilon$-duality-gap solution for SCSC or an $\epsilon$-stationary solution for WCSC min-max problems. We focus on comparison of existing results without assuming smoothness of the objective function. Restriction means whether an additional condition about the objective function's structure is imposed.

| Setting | Works | Restriction | Convergence | Complexity |
|---|---|---|---|---|
| SCSC | [32] | No | Duality Gap | $O\left(1/\epsilon^2\right)$ |
|  | [41] | Yes | Primal Gap | $O\left(1/\epsilon + n\log(1/\epsilon)\right)$ |
|  | **This paper** | **No** | **Duality Gap** | $\boldsymbol{O\left(1/\epsilon\right)}$ |
| WCSC | [36] | No | Nearly Stationary | $\widetilde{O}\left(1/\epsilon^6\right)$ |
|  | [36] | Yes | Nearly Stationary | $\widetilde{O}\left(1/\epsilon^4 + n/\epsilon^2\right)$ |
|  | **This paper** | **No** | **Nearly Stationary** | $\boldsymbol{\widetilde{O}\left(1/\epsilon^4\right)}$ |

nearly optimal complexity for solving WCSC problems without the smoothness assumption on the objective [2] [1]. Finally, we summarize our results and the comparison with existing results in Table 1.

## 2   Related Work

Below, we provide an overview of related results in this area and the review is not necessarily exhaustive. In addition, we focus on the stochastic algorithms, and leave deterministic algorithms [4, 33, 42, 12, 34, 19, 14, 20, 28, 15] out of our discussion.

[32] is one of the early works that studies stochastic primal-dual gradient methods for convex-concave min-max problems, which establishes a convergence rate of $O(1/\sqrt{T})$ for the duality gap of general convex-concave problems. Following this work, many studies have tried to improve the algorithm and the analysis for a certain class of problems by exploring the smoothness condition of some component functions [23, 47, 21] or bilinear structure of the objective function [5, 6]. For example, [47] considers a family of min-max problems whose objective is $f(x) + g(x) + \phi(x,y) - J(y)$, where the smoothness condition is imposed on $f$ and $\phi$ and strong convexity is imposed on $f$ if necessary, and establishes optimal or nearly optimal complexity of a stochastic primal-dual hybrid algorithm. Although the dependence on each problem parameter of interest is made (nearly) optimal, the worst case complexity is still $O(1/\sqrt{T})$. [21] considers single-call stochastic extra-gradient and establishes $O(1/T)$ rate for smooth and strongly monotone variational inequalities in terms of the square distance from the returned solution to the saddle point. [44] also considers variational inequalities with a smoothing technique, so that it handles nonsmooth problems, but they derive the convergence of the square distance from the returned solution to the saddle point, as in [21]. The present work is complementary to these developments by making no assumption on smoothness or the structure of the objective but considers strong (weak) convexity and strong concavity of the objective function. It has applications in robust learning with non-smooth loss functions [41, 36].

In the machine learning community, many works have considered stochastic primal-dual algorithms for solving regularized loss minimization problems, whose min-max formulation usually exhibits bi-linear structure [46, 37, 39, 10, 35]. For example, [46] designs a stochastic primal-dual coordinate (SPDC) method for SCSC problems with bilinear structure, which enjoys a linear convergence for the duality gap. Similarly, in [45, 38], different variants of SPDC are proposed and analyzed for problems with the bilinear structure. [35] proposes stochastic variance reduction methods for a family of saddle-point problems with special structure that yields a linear convergence rate. An exception that makes no smoothness assumption and imposes no bilinear structure is a recent work [41]. It considers a family of functions $f(x,y) = y^\top \ell(x) - \phi^*(y) + g(x)$ and proposes a stochastic primal-dual algorithm similar to Epoch-GDA. The key difference is that [41] designs a particular scheme that computes a restarting dual solution based on $\nabla\phi(\ell(\bar{x}))$, where $\bar{x}$ is a restarting primal solution in order to derive a fast rate of $O(1/T)$ under strong convexity and strong concavity. Additionally, their

fast rate $O(1/T)$ is in terms of the primal objective gap, which is weaker than our convergence result in terms of the duality gap.

There is also increasing interest in stochastic primal-dual algorithms for solving WCSC min-max problems. To the best of our knowledge, [36] is probably the first work that comprehensively studies stochastic algorithms for solving WCSC min-max problems. To find a nearly $\epsilon$-stationary point, their algorithms suffer from an $O(1/\epsilon^6)$ iteration complexity without strong concavity and an $O(1/\epsilon^4 + n/\epsilon^2)$ complexity with strong concavity and a special structure of the objective function that is similar to that imposed in [41]. Some recent works are trying to improve the complexity for solving WCSC min-max problems by exploring other conditions (e.g., smoothness) [25, 29, 40, 22]. For example, [25] establishes an $O(1/\epsilon^4)$ complexity for a single-loop stochastic gradient descent ascent method, while [29, 40, 22] make use of variance reduction or momentum to achieve $O(1/\epsilon^3)$ complexity. However, their analysis requires the smoothness condition and some of their algorithms need to use a large mini-batch size in the order $O(1/\epsilon^2)$. In contrast, we impose neither assumption about smoothness nor special structure of the objective function. The complexity of our algorithm is $\widetilde{O}(1/\epsilon^4)$ for finding a nearly $\epsilon$-stationary point, which is the state of the art result for the considered non-smooth WCSC problem.

# 3   Preliminaries

This section provides some notations and assumptions used in the paper. We let $\|\cdot\|$ denote the Euclidean norm of a vector. Given a function $f : \mathbb{R}^d \to \mathbb{R}$, we denote the Fréchet subgradients and limiting Fréchet gradients by $\hat{\partial} f$ and $\partial f$, respectively, i.e., at $x$, $\hat{\partial} f(x) = \{v \in \mathbb{R}^d : \lim_{x \to x'} \inf \frac{f(x) - f(x') - v^\top (x - x')}{\|x - x'\|} \geq 0\}$, and $\partial f(x) = \{v \in \mathbb{R}^d : \exists x_k \xrightarrow{f} x, v_k \in \hat{\partial} f(x_k), v_k \to v, v \in \hat{\partial} f(x)\}$. Here $x_k \xrightarrow{f} x$ represents $x_k \to x$ with $f(x_k) \to f(x)$. A function $f(x)$ is $\mu$-strongly convex on $X$ if for any $x, x' \in X$, $\partial f(x')^\top (x - x') + \frac{\mu}{2}\|x - x'\|^2 \leq f(x) - f(x')$. A function $f(x)$ is $\rho$-weakly convex on $X$ for any $x, x' \in X$ $\partial f(x')^\top (x - x') - \frac{\rho}{2}\|x - x'\|^2 \leq f(x) - f(x')$. Let $\mathcal{G}_x \in \partial_x f(x, y; \xi)$ denote a stochastic subgradient of $f$ at $x$ given $y$, where $\xi$ is used to denote the random variable. Similarly, let $\mathcal{G}_y \in \partial_y f(x, y; \xi)$ denote a stochastic sugradient of $f$ at $y$ given $x$. Let $\Pi_\Omega[\cdot]$ denote the projection onto the set $\Omega$, and let $\mathcal{B}(x, R)$ denote an Euclidean ball centered at $x$ with a radius $R$. Denote by $dist(x, X)$ the distance between $x$ and the set $X$, i.e., $dist(x, X) = \min_{v \in X} \|x - v\|$. Let $\tilde{O}(\cdot)$ hide some logarithmic factors.

For a WCSC min-max problem, it is generally a hard problem to find a saddle point. Hence, we use *nearly $\epsilon$-stationarity* as the measure of convergence for solving WCSC problems [36], which is defined as follows.

**Definition 1.** *A solution $x$ is a nearly $\epsilon$-stationary point of $\min_x \psi(x)$ if there exist $z$ and a constant $c > 0$ such that $\|z - x\| \leq c\epsilon$ and $dist(0, \partial\psi(z)) \leq \epsilon$.*

For a $\rho$-weakly convex function $\psi(x)$, let $z = \arg\min_{x \in \mathbb{R}^d} \psi(x) + \frac{\gamma}{2}\|x - \tilde{x}\|^2$ where $\gamma > \rho$ and $\tilde{x} \in \mathbb{R}^d$ is a reference point. Due to the strong convexity of the above problem, $z$ is unique and $0 \in \partial\psi(z) + \gamma(z - \tilde{x})$, which results in $\gamma(\tilde{x} - z) \in \partial\psi(z)$, so that $dist(0, \partial\psi(z)) \leq \gamma\|\tilde{x} - z\|$. According to [8, 7, 9], we can find a nearly $\epsilon$-stationary point $\tilde{x}$ as long as $\gamma\|\tilde{x} - z\| \leq \epsilon$.

Before ending this section, we present some assumptions that will be imposed in our analysis.

**Assumption 1.** *$X$ and $Y$ are closed convex sets. There exist initial solutions $x_0 \in X, y_0 \in Y$ and $\epsilon_0 > 0$ such that $Gap(x_0, y_0) \leq \epsilon_0$.*

**Assumption 2.** *(1) $f(x, y)$ is $\mu$-strongly convex in $x$ for any $y \in Y$ and $\lambda$-strongly concave in $y$ for any $x \in X$. (2) There exist $B_1, B_2 > 0$ such that $\mathrm{E}[\exp(\frac{\|\mathcal{G}_x\|^2}{B_1^2})] \leq \exp(1)$ and $\mathrm{E}[\exp(\frac{\|\mathcal{G}_y\|^2}{B_2^2})] \leq \exp(1)$.*

**Assumption 3.** *(1) $f(x, y)$ is $\rho$-weakly convex in $x$ for any $y \in Y$ and is $\lambda$-strongly concave in $y$ for any $x \in X$. (2) $\mathrm{E}[\|\mathcal{G}_x\|^2] \leq M_1^2$ and $\mathrm{E}[\|\mathcal{G}_y\|^2] \leq M_2^2$.*

**Remark:** When $f(x, y)$ is smooth in $x$ and $y$, the second condition in the above assumption can be replaced by the bounded variance condition.

---
**Algorithm 1** Epoch-GDA for SCSC Min-Max Problems
---
1: Init.: $x_0^1 = x_0 \in X, y_0^1 = y_0 \in Y, \eta_x^1, \eta_y^1, R_1, T_1$
2: **for** $k = 1, 2, ..., K$ **do**
3:     **for** $t = 0, 1, 2, ..., T_k - 1$ **do**
4:         Compute stochastic gradients $\mathcal{G}_{x,t}^k \in \partial_x f(x_t^k, y_t^k; \xi_t^k)$ and $\mathcal{G}_{y,t}^k \in \partial_y f(x_t^k, y_t^k; \xi_t^k)$.
5:         $x_{t+1}^k = \Pi_{X \cap \mathcal{B}(x_0^k, R_k)}(x_t^k - \eta_x^k \mathcal{G}_{x,t}^k)$
6:         $y_{t+1}^k = \Pi_{Y \cap \mathcal{B}(y_0^k, R_k)}(y_t^k + \eta_y^k \mathcal{G}_{y,t}^k)$
7:     **end for**
8:     $x_0^{k+1} = \bar{x}_k = \frac{1}{T_k} \sum_{t=0}^{T_k-1} x_t^k, y_0^{k+1} = \bar{y}_k = \frac{1}{T_k} \sum_{t=0}^{T_k-1} y_t^k$
9:     $\eta_x^{k+1} = \frac{\eta_x^k}{2}, \eta_y^{k+1} = \frac{\eta_y^k}{2}, R_{k+1} = R_k/\sqrt{2}, T_{k+1} = 2T_k$.
10: **end for**
11: Return $(\bar{x}_K, \bar{y}_K)$.
---

## 4 Main Results

### 4.1 Strongly-Convex Strongly-Concave Min-Max Problems

In this subsection, we present the main result for solving SCSC problems. The proposed Epoch-GDA algorithm for SCSC min-max problems is shown in Algorithm 1. As illustrated, our algorithm consists of a series of epochs. In each epoch (Line 3 to 7), standard primal-dual updates are performed. After an epoch ends, in Line 8, the solutions $\bar{x}_k$ and $\bar{y}_k$ averaged over the epoch are returned as the initialization for the next epoch. In Line 9, step sizes $\eta_{x,k+1}$ and $\eta_{y,k+1}$, the radius $R_{k+1}$ and the number of iterations $T_{k+1}$ are also adjusted for the next epoch. The ball constraints $\mathcal{B}(x_0^k, R_k)$ and $\mathcal{B}(y_0^k, R_k)$ at each iteration are used for the convergence analysis in high probability as in [16, 17]. It is clear that Epoch-GDA can be considered as a primal-dual variant of Epoch-GD [16, 17].

The following theorem shows that the iteration complexity of Algorithm 1 to achieve an $\epsilon$-duality gap for a general SCSC problem (1) is $O(1/\epsilon)$.

**Theorem 1.** *Suppose Assumption 1 and Assumption 2 hold and let $\delta \in (0, 1)$ be a failing probability and $\epsilon \in (0, 1)$ be the target accuracy level for the duality gap. Let $K = \lceil \log(\frac{\epsilon_0}{\epsilon}) \rceil$ and $\tilde{\delta} = \delta/K$, and the initial parameters are set by $R_1 \geq 2\sqrt{\frac{2\epsilon_0}{\min\{\mu,\lambda\}}}$, $\eta_x^1 = \frac{\min\{\mu,\lambda\}R_1^2}{40(5+3\log(1/\tilde{\delta}))B_1^2}$, $\eta_y^1 = \frac{\min\{\mu,\lambda\}R_1^2}{40(5+3\log(1/\tilde{\delta}))B_2^2}$ and*

$$T_1 \geq \frac{\max\left\{320^2(B_1 + B_2)^2 3\log(1/\tilde{\delta}), 3200(5 + 3\log(1/\tilde{\delta}))\max\{B_1^2, B_2^2\}\right\}}{\min\{\mu,\lambda\}^2 R_1^2}.$$

*Then the total number of iterations of Algorithm 1 to achieve an $\epsilon$-duality gap, i.e., $Gap(\bar{x}_K, \bar{y}_K) \leq \epsilon$, with probability $1 - \delta$ is*

$$T_{tot} = \frac{\max\left\{320^2(B_1 + B_2)^2 3\log(1/\tilde{\delta}), 3200(5 + 3\log(1/\tilde{\delta}))\max\{B_1^2, B_2^2\}\right\}}{4\min\{\mu,\lambda\}\epsilon}.$$

**Remark 1.** *To the best of our knowledge, this is the first study that achieves a fast rate of $O(1/T)$ for the duality gap of a general SCSC min-max problem without any special structure assumption or smoothness of the objective function and an additional computational cost. In contrast, even if the algorithm in [41] attains the $O(1/T)$ rate of convergence, it i) only guarantees the convergence of the primal objective gap, rather than the duality gap, ii) additionally requires a special structure of the objective function, and iii) needs an extra $O(n)$ computational cost of the deterministic update at each outer loop to handle the maximization over $y$. In contrast, Algorithm 1 has stronger theoretical results with less restrictions of the problem structures and computational cost.*

**Remark 2.** *A lower bound of $O(1/T)$ for stochastic strongly convex minimization problems has been proven in [1, 17]. Due to $Gap(x, y) \geq P(x) - P(x^*)$, bounding the duality gap is more difficult than bounding the primal gap. This means that our convergence rate matches the lower bound and is therefore the best possible convergence rate without adding more assumptions.*

### 4.2 Weakly-Convex Strongly-Concave Problems

---

**Algorithm 2** Epoch-GDA for WCSC Min-Max Problems

---
1: Init.: $x_0^1 = x_0 \in X, y_0^1 = y_0 \in Y, \gamma = 2\rho$.
2: **for** $k = 1, 2, ..., K$ **do**
3:     Set $T_k = \frac{106(k+1)}{3}, \eta_x^k = \frac{4}{\rho(k+1)}, \eta_y^k = \frac{2}{\lambda(k+1)}$.
4:     **for** $t = 1, 2, ..., T_k$ **do**
5:         Compute $\mathcal{G}_{x,t}^k \in \partial_x f(x_t^k, y_t^k; \xi_t^k)$ and $\mathcal{G}_{y,t}^k \in \partial_y f(x_t^k, y_t^k; \xi_t^k)$.
6:         $x_{t+1}^k = \arg\min_{x \in X} x^\top \mathcal{G}_{x,t}^k + \frac{1}{2\eta_x^k}\|x - x_t^k\|^2 + \frac{\gamma}{2}\|x - x_0^k\|^2$
7:         $y_{t+1}^k = \arg\min_{y \in Y} -y^\top \mathcal{G}_{y,t}^k + \frac{1}{2\eta_y^k}\|y - y_t^k\|^2$
8:     **end for**
9:     $x_0^{k+1} = \bar{x}_k = \frac{1}{T}\sum_{t=0}^{T-1} x_t^k, y_0^{k+1} = \bar{y}_k = \frac{1}{T}\sum_{n=0}^{T-1} y_t^k$
10: **end for**
11: Return $x_0^\tau$ by $\tau$ randomly sampled from $\{1, ..., K\}$.

---

In this subsection, we present the convergence results for solving WCSC problems, where the objective function $f(x, y)$ in (1) is $\rho$-weakly convex in $x$ and $\lambda$-strongly concave in $y$. The proposed Epoch-GDA algorithm for WCSC min-max problems is summarized in Algorithm 2. As our Algorithm 1, Algorithm 2 consists of a number of epochs. As shown in Line 4 to Line 8, each epoch performs primal-dual updates on $x$ and $y$. When updating $x$ at the $k$-th stage, an additional regularizer $\frac{\gamma}{2}\|x - x_0^k\|^2$ is added, where the value $\gamma = 2\rho$. The added term is used to handle the weak convexity condition. After an epoch ends, average solutions of both $x$ and $y$ are restarted as the initial ones for the next epoch. The step sizes for updating $x$ and $y$ are set to $O(1/(\rho k))$ and $O(1/(\lambda k))$ at the $k$-th epoch, respectively. If we define $\hat{f}_k(x, y) = f(x, y) + \frac{\gamma}{2}\|x - x_0^k\|^2$, we can see that $\hat{f}_k(x, y)$ is $\rho$-strongly convex in $x$ and $\lambda$-strongly concave in $y$, since $f(x, y)$ is $\rho$-weakly convex and $\gamma = 2\rho$. Indeed, for each inner loop of Algorithm 2, we actually work on the SCSC problem $\min_{x \in X} \max_{y \in Y} \hat{f}_k(x, y)$.

It is worth mentioning the key difference between our algorithm and the recently proposed stochastic algorithm PG-SMD [36] for WCSC problems with a special structural objective function. PG-SMD also consists of two loops. For each inner loop, it runs the same updates with the added regularizer on $x$ as Algorithm 2. It restarts $x$ by averaging the solutions over the inner loop, like our $\bar{x}_k$, but restarts $y$ by taking the deterministic maximization of (1) over $y$ given $\bar{x}_k$, leading to an additional $O(n)$ computational complexity per epoch. In addition, PG-SMD sets $\eta_y^k = O(1/(\gamma\lambda^2 k))$. Although Algorithm 2 shares similar updates to PG-SMD, our analysis yields stronger results under weaker assumptions — the same iteration complexity $\tilde{O}(1/\epsilon^4)$ without deterministic updates for $y$ and special structure in the objective function. This is due to our sharper analysis that makes use of the telescoping sum based on the duality gap of $\hat{f}_k$ instead of the primal objective gap.

Let $\hat{P}(x) = P(x) + \mathbb{I}_X(x)$ where $\mathbb{I}_X(x)$ denotes the indicator function of the constraint set $X$ at $x$. The convergence result of Algorithm 2 that achieves a nearly $\epsilon$-stationary point with $\tilde{O}(1/\epsilon^4)$ iteration complexity is summarized below.

**Theorem 2.** *Suppose Assumption 3 holds. Algorithm 2 guarantees* $\mathrm{E}[dist(0, \partial \hat{P}(\hat{x}_\tau^*))^2] \leq$ $\gamma^2 \mathrm{E}[\|\hat{x}_\tau^* - x_0^\tau\|^2] \leq \epsilon^2$ *after* $K = \max\left\{ \frac{1696\gamma(\frac{2M_1^2}{\rho} + \frac{M_2^2}{\lambda})}{\epsilon^2} \ln(\frac{1696\gamma(\frac{2M_1^2}{\rho} + \frac{M_2^2}{\lambda})}{\epsilon^2}), \frac{1376\gamma\epsilon_0}{5\epsilon^2} \right\}$ *epochs, where $\tau$ is randomly sampled from $\{1, ..., K\}$ and $(\hat{x}_k^*, \hat{y}_k^*)$ is the saddle-point of $\hat{f}_k(x, y)$. The total number of iteration is $\sum_{k=1}^K T_k = \tilde{O}(\frac{1}{\epsilon^4})$.*

**Remark 3.** *Theorem 2 shows that the iteration complexity of Algorithm 2 to attain an $\epsilon$-nearly stationary point is $\tilde{O}(1/\epsilon^4)$. It improves the result of [36] for WCSC problems in terms of two aspects. First, [36] requires a stronger condition on the structure of the objective function, while our analysis simply assumes a general objective function $f(x, y)$. Second, [36] requires to solve the maximization over $y$ at each epoch, which may introduce an $O(n)$ computational complexity for $y \in \mathbb{R}^n$ [2]. In contrast, our algorithm restarts both the primal variable $x$ and dual variable $y$ at each epoch, which does not need an additional cost.*

*Finally, we note that when $f(x, y)$ is smooth in $x$ and $y$, we can use stochastic Mirror Prox algorithm [23] to replace the stochastic gradient descent ascent updates (Step 6 and Step 7) such that we can use a bounded variance assumption of the stochastic gradients instead of bounded second-order moments. It is a simple exercise to finish the proof by following our analysis of Theorem 2.*

We prove the expectation result for WCSC in Theorem 2 for consistency with previous results [36]. In fact, we can also derive the high probability version. We provide a proof sketch at the end of the proof of Theorem 2 in the appendix and leave the details to the longer version.

## 5 Analysis

In this section, we present the proof of Theorem 1 and a proof sketch of Theorem 2. As we mentioned at the introduction, the key challenge in the analysis of Epoch-GDA lies in handling the variable distance measure $\|\hat{x}(y_1) - x_0\|^2 + \|\hat{y}(x_1) - y_0\|^2$ for any $(x_0, y_0) \in X \times Y$ and $(x_1, y_1) \in X \times Y$ and its connection to the duality gaps, where $\hat{x}(y_1) = \arg\min_{x' \in X} f(x', y_1)$ and $\hat{y}(x_1) = \arg\max_{y' \in Y} f(x_1, y')$. Hence, we first introduce the following key lemma that is useful in the analysis of Epoch-GDA for both SCSC and WCSC problems. It connects the variable distance measure $\|\hat{x}(y_1) - x_0\|^2 + \|\hat{y}(x_1) - y_0\|^2$ to the duality gaps at $(x_0, y_0)$ and $(x_1, y_1)$.

**Lemma 1.** *Consider the following $\mu$-strongly convex in $x$ and $\lambda$-strongly concave problem $\min_{x \in \Omega_1} \max_{y \in \Omega_2} f(x, y)$. Let $(x^*, y^*)$ denote the saddle point solution to this problem. Suppose we have two solutions $(x_0, y_0) \in \Omega_1 \times \Omega_2$ and $(x_1, y_1) \in \Omega_1 \times \Omega_2$. Then the following relation between variable distance and duality gaps holds*

$$\frac{\mu}{4}\|\hat{x}(y_1) - x_0\|^2 + \frac{\lambda}{4}\|\hat{y}(x_1) - y_0\|^2 \leq \max_{y' \in \Omega_2} f(x_0, y') - \min_{x' \in \Omega_1} f(x', y_0)$$
$$+ \max_{y' \in \Omega_2} f(x_1, y') - \min_{x' \in \Omega_1} f(x', y_1). \quad (2)$$

### 5.1 Proof of Theorem 1 for the SCSC setting

The key idea is to first show the convergence of the duality gap with respect to the ball constraints $\mathcal{B}(x_0^k, R_k)$ and $\mathcal{B}(y_0^k, R_k)$ in an epoch (Lemma 2). Then we investigate the condition to make $\hat{x}(\bar{y}_k) \in \mathcal{B}(x_0^k, R_k)$ and $\hat{y}(\bar{x}_k) \in \mathcal{B}(y_0^k, R_k)$ given the average solution $(\bar{x}_k, \bar{y}_k)$, which allows us to derive the duality gap $\text{Gap}(\bar{x}_k, \bar{y}_k)$ for the original problem. Finally, under such conditions, we show how the duality gap between two consecutive outer loops can be halved (Theorem 3), which implies the total iteration complexity (Theorem 1). Below, we omit superscript $k$ when it applies to all epochs.

**Lemma 2.** *Suppose Assumption 2 holds. Let Line 3 to 7 of Algorithm 1 run for $T$ iterations (omitting the $k$-index) by fixed step sizes $\eta_x$ and $\eta_y$. Then with the probability at least $1 - \tilde{\delta}$ where $0 < \tilde{\delta} < 1$, for any $x \in X \cap \mathcal{B}(x_0, R)$ and $y \in Y \cap \mathcal{B}(y_0, R)$, $\bar{x} = \sum_{t=0}^{T-1} x_t/T$, $\bar{y} = \sum_{t=0}^{T-1} y_t/T$ satisfy*

$$f(\bar{x}, y) - f(x, \bar{y}) \leq \frac{\|x - x_0\|^2}{\eta_x T} + \frac{\|y - y_0\|^2}{\eta_y T} + \frac{\eta_x B_1^2 + \eta_y B_2^2}{2}(5 + 3\log(1/\tilde{\delta}))$$
$$+ \frac{4(B_1 + B_2)R\sqrt{3\log(1/\tilde{\delta})}}{\sqrt{T}}. \quad (3)$$

**Remark 4.** *Lemma 2 is a standard analysis for an epoch of Algorithm 1. The difficulty arises when attempting to plug $x$ and $y$ into (3). In order to derive the duality gap on the LHS of (3), we have to plug in $x \leftarrow \hat{x}(\bar{y})$ and $y \leftarrow \hat{y}(\bar{x})$. Nevertheless, it is unclear whether $\hat{x}(\bar{y}) \in \mathcal{B}(x_0, R)$ and $\hat{y}(\bar{x}) \in \mathcal{B}(y_0, R)$, which is the requirement for $x$ and $y$ to be plugged into (3). In the following lemma, we investigate the condition to make $\hat{x}(\bar{y}) \in \mathcal{B}(x_0, R)$ and $\hat{y}(\bar{x}) \in \mathcal{B}(y_0, R)$ based on Lemma 1.*

**Lemma 3.** *Suppose Assumption 2 holds. Let $\hat{x}_R(y) := \arg\min_{x \in X \cap \mathcal{B}(x_0, R)} f(x, y)$ and $\hat{y}_R(x) := \arg\max_{y \in Y \cap \mathcal{B}(y_0, R)} f(x, y)$. Assume the initial duality gap $\text{Gap}(x_0, y_0) \leq \epsilon_0$. Let Lines 3 to 7 of Algorithm 1 run $T$ iterations with $\tilde{\delta} \in (0, 1)$, $R \geq 2\sqrt{\frac{2\epsilon_0}{\min\{\mu, \lambda\}}}$, $\eta_x = \frac{\min\{\mu, \lambda\}R^2}{40(5 + 3\log(1/\tilde{\delta}))B_1^2}$,*

$\eta_y = \frac{\min\{\mu,\lambda\}R^2}{40(5+3\log(1/\tilde{\delta}))B_2^2}$ and

$$T \geq \frac{\max\left\{320^2(B_1+B_2)^2 3\log(1/\tilde{\delta}), 3200(5+3\log(1/\tilde{\delta}))\max\{B_1^2, B_2^2\}\right\}}{\mu^2 R^2}.$$

*Then, with probability at least $1 - \tilde{\delta}$, it holds $\|\hat{x}_R(\bar{y}) - x_0\| < R$, $\|\hat{y}_R(\bar{x}) - y_0\| < R$.*

**Remark 5.** *Lemma 3 shows that if we properly set the values of $R$, $\eta_x$, $\eta_y$ and $T$, then $\hat{x}_R(\bar{y})$ and $\hat{y}_R(\bar{x})$ are the interior points of $\mathcal{B}(x_0, R)$ and $\mathcal{B}(y_0, R)$ with high probability. Therefore, we conclude that $\hat{x}(\bar{y}) = \hat{x}_R(\bar{y})$ and $\hat{y}(\bar{x}) = \hat{y}_R(\bar{x})$ with probability $1 - \tilde{\delta}$ under the conditions of Lemma 3, which allows us to derive the duality gap in LHS of (3) of Lemma 2.*

We would highlight that $\hat{x}(\bar{y}) \in \mathcal{B}(x_0, R)$ and $\hat{y}(\bar{x}) \in \mathcal{B}(y_0, R)$ have to be confirmed in high probability, rather than in expectation. If we show $\mathrm{E}[\|\hat{x}(\bar{y}) - x_0\|] < R$, it is still unclear $\hat{x}(\bar{y}) \in \mathcal{B}(x_0, R)$, as pointed in [47]. The following theorem gives the relation of duality gaps between two consecutive epochs of Algorithm 1 by using Lemma 2 and the conditions proven by Lemma 3.

**Theorem 3.** *Consider the $k$-th epoch of Algorithm 1 with an initial solution $(x_0^k, y_0^k)$ and the ending averaged solution $(\bar{x}_k, \bar{y}_k)$. Suppose Assumption 2 holds and $Gap(x_0^k, y_0^k) \leq \epsilon_{k-1}$. Let $R_k \geq 2\sqrt{\frac{2\epsilon_{k-1}}{\min\{\mu,\lambda\}}}$ (i.e. $\epsilon_{k-1} \leq \frac{\min\{\mu,\lambda\}R_k^2}{8}$), $\eta_x^k = \frac{\min\{\mu,\lambda\}R_k^2}{40(5+3\log(1/\tilde{\delta}))B_1^2}$, $\eta_y^k = \frac{\min\{\mu,\lambda\}R_k^2}{40(5+3\log(1/\tilde{\delta}))B_2^2}$ and*

$$T_k \geq \frac{\max\left\{320^2(B_1+B_2)^2 3\log(1/\tilde{\delta}), 3200(5+3\log(1/\tilde{\delta}))\max\{B_1^2, B_2^2\}\right\}}{\min\{\mu,\lambda\}^2 R_k^2}.$$

*Then we have with probability $1 - \tilde{\delta}$, $Gap(\bar{x}_k, \bar{y}_k) \leq \frac{\min\{\mu,\lambda\}R_k^2}{16}$.*

**Remark 6.** *Theorem 3 shows that after running $T_k$ iterations at the $k$-th stage, the upper bound of the duality gap would be halved with high probability, i.e., from $\frac{\min\{\mu,\lambda\}R_k^2}{8}$ to $\frac{\min\{\mu,\lambda\}R_k^2}{16}$. Then, in order to make the duality gap of each outer loop of Algorithm 1 halved from the last epoch, we can simply set $R_{k+1}^2 = \frac{R_k^2}{2}$, and accordingly, $\eta_{x,k+1} = \frac{\eta_x^k}{2}$, $\eta_{y,k+1} = \frac{\eta_y^k}{2}$ and $T_{k+1} = 2T_k$.*

*Proof.* (of Theorem 3) For any $x \in \mathcal{B}(x_0^k, R_k)$ and $y \in \mathcal{B}(y_0^k, R_k)$, we have $\|x - x_0^k\| \leq R$ and $\|y - y_0^k\| \leq R$, so by (3) of Lemma 2, we have with probability $1 - \tilde{\delta}$

$$f(\bar{x}_k, y) - f(x, \bar{y}_k) \overset{(a)}{\leq} \frac{R_k^2}{\eta_x^k T_k} + \frac{R_k^2}{\eta_y^k T_k} + \frac{\eta_x^k B_1^2}{2}(5 + 3\log(1/\tilde{\delta})) + \frac{\eta_y^k B_2^2}{2}(5 + 3\log(1/\tilde{\delta}))$$

$$+ \frac{4(B_1 + B_2)R_k\sqrt{3\log(1/\tilde{\delta})}}{\sqrt{T_k}} \overset{(b)}{\leq} \frac{\min\{\mu,\lambda\}R_k^2}{16}, \tag{4}$$

where inequality $(a)$ is due to $x \in \mathcal{B}(x_0^k, R_k)$ and $y \in \mathcal{B}(y_0^k, R_k)$. Inequality $(b)$ is due to the values of $\eta_x^k$, $\eta_y^k$ and $T_k$. Recall the definitions $\hat{x}(\bar{y}_k) = \arg\min_{x \in X} f(x, \bar{y}_k)$ and $\hat{y}(\bar{x}_k) = \arg\max_{y \in Y} f(\bar{x}_k, y)$. By Lemma 3, we have $\hat{x}(\bar{y}_k) \in \mathcal{B}(x_0^k, R_k)$ and $\hat{y}(\bar{x}_k) \in \mathcal{B}(y_0^k, R_k)$ with probability $1 - \tilde{\delta}$. Then from (4) we have

$$Gap(\bar{x}_k, \bar{y}_k) = \max_{y \in Y} f(\bar{x}_k, y) - \min_{x \in X} f(x, \bar{y}_k) \leq \frac{\min\{\mu,\lambda\}R_k^2}{16}.$$

$\square$

Given the condition $Gap(x_0^k, y_0^k) \leq \epsilon_{k-1} \leq \frac{\min\{\mu,\lambda\}R_k^2}{8}$, we then conclude that running $T_k$ iterations in an epoch of Algorithm 1 would halve the duality gap with high probability. As indicated in Theorem 3, the duality gap $Gap(\bar{x}_k, \bar{y}_k)$ can be halved as long as the condition of Theorem 3 holds. Then Theorem 1 is implied (the detailed proof is in Supplementary Materials).

## 5.2 Proof Sketch of Theorem 2 for the WCSC setting

Due to limit of space, we only present a sketch here and present the full proof in the Supplement. Recall $\hat{f}_k(x, y) = f(x, y) + \frac{\gamma}{2}\|x - x_0^k\|^2$. Let us denote its duality gap by $\widehat{Gap}_k(x, y) = \hat{f}_k(x, \hat{y}_k(x)) - \hat{f}_k(\hat{x}_k(y), y)$, where we define $\hat{y}_k(x) := \arg\max_{y' \in Y} \hat{f}_k(x, y')$ given $x \in X$ and

$\hat{x}_k(y) := \arg\min_{x' \in X} \hat{f}_k(x', y)$ given $y \in Y$. Its saddle point solution is denoted by $(\hat{x}_k^*, \hat{y}_k^*)$, i.e., $\hat{f}_k(\hat{x}_k^*, y) \leq \hat{f}_k(\hat{x}_k^*, \hat{y}_k^*) \leq \hat{f}_k(x, \hat{y}_k^*)$ for any $x \in X$ and $y \in Y$. The key idea of our analysis is to connect the duality gap $\widehat{\text{Gap}}_k(x_0^k, y_0^k)$ to $\gamma^2 \|\hat{x}_k^* - x_0^k\|^2$, and then by making $\gamma^2 \|\hat{x}_k^* - x_0^k\|^2 \leq \epsilon^2$, we can show that $x_0^k$ is a nearly $\epsilon$-stationary point. To this end we first establish a bound of the duality gap for the regularized problem $\hat{f}_k(x, y)$ for the $k$-th epoch (Lemma 4). Then we connect it to $\gamma\|\hat{x}_k^* - x_0^k\|^2$ (Lemma 5). Finally, we bound $\gamma\|\hat{x}_k^* - x_0^k\|^2$ by a telescoping sum of $\text{E}[\widehat{\text{Gap}}_k(x_0^k, y_0^k)] - \text{E}[\widehat{\text{Gap}}_{k+1}(x_0^{k+1}, y_0^{k+1})]$ and $\text{E}[P(x_0^k) - P(x_0^{k+1})]$.

## 6 Conclusions

In this paper, we filled the gap between stochastic min-max and minimization optimization problems. We proposed Epoch-GDA algorithms for general SCSC and general WCSC problems, which do not impose any additional assumptions on the smoothness or the structure of the objective function. Our key lemma provides sharp analysis of Epoch-GDA for both problems. For SCSC min-max problems, to the best of our knowledge, our result is the first one to show that Epoch-GDA achieves the optimal rate of $O(1/T)$ for the duality gap of general SCSC min-max problems. For WCSC min-max problems, our analysis allows us to derive the best complexity $\tilde{O}(1/\epsilon^4)$ of Epoch-GDA to reach a nearly $\epsilon$-stationary point, which does not require smoothness, large mini-batch sizes or other structural conditions.

## Broader Impact

A discussion about broader impact is not applicable since our work is very theoretical and currently has no particular application.

## Acknowledgments and Disclosure of Funding

T. Yang is partially supported by National Science Foundation Career Award (NSF 1844403) and NSF grant #1933212. Most work of Y. Yan was done when he worked in the University of Iowa.

## Footnotes

[1]Although [2] only concerns the lower bound of finding a stationary point of smooth non-convex problems $\min_x f(x)$ through stochastic first-order oracle, it is a special case of the WCSC problem.

[2]Although the exact maximization over $y$ for restarting next epoch might be solved approximately, it still requires additional overhead.

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
