[Supplementary Material]

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

 $\mathrm{E}[\widehat{\mathrm{Gap}}_k(x_0^k, y_0^k)] - \mathrm{E}[\widehat{\mathrm{Gap}}_{k+1}(x_0^{k+1}, y_0^{k+1})]$ and $\mathrm{E}[P(x_0^k) - P(x_0^{k+1})]$.

## 6 Conclusions

In this paper, we filled the gap between stochastic min-max and minimization optimization problems. We proposed Epoch-GDA algorithms for general SCSC and general WCSC problems, which do not impose any additional assumptions on the smoothness or the structure of the objective function. Our key lemma provides sharp analysis of Epoch-GDA for both problems. For SCSC min-max problems, to the best of our knowledge, our result is the first one to show that Epoch-GDA achieves the optimal rate of $O(1/T)$ for the duality gap of general SCSC min-max problems. For WCSC min-max problems, our analysis allows us to derive the best complexity $\tilde{O}(1/\epsilon^4)$ of Epoch-GDA to reach a nearly $\epsilon$-stationary point, which does not require smoothness, large mini-batch sizes or other structural conditions.

## Broader Impact

A discussion about broader impact is not applicable since our work is very theoretical and currently has no particular application.

## Acknowledgments and Disclosure of Funding

T. Yang is partially supported by National Science Foundation Career Award (NSF 1844403) and NSF grant #1933212. Most work of Y. Yan was done when he worked in the University of Iowa.

## Footnotes

[1] Although [2] only concerns the lower bound of finding a stationary point of smooth non-convex problems $\min_x f(x)$ through stochastic first-order oracle, it is a special case of the WCSC problem.

[2] Although the exact maximization over $y$ for restarting next epoch might be solved approximately, it still requires additional overhead.

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

## A  Proof of Theorem 2 for the WCSC setting

For the proof of Theorem 2, we need to merge the constraint set $X$ into the objective function, so that we can derive the convergence of the near $\epsilon$-stationary point. Recall originally $\hat{f}_k(x,y) = f(x,y) + \frac{\gamma}{2}\|x - x_0^k\|^2$. Now we re-define $\hat{f}_k(x,y) = f(x,y) + \frac{\gamma}{2}\|x - x_0^k\|^2 + \mathbb{I}_X(x)$, where $I_X(x)$ is the indicator function of the constraint set $X$ at $x$. Similarly, we re-define $P(x) = \max_{y \in Y} f(x,y) + \mathbb{I}_X(x)$ by merging the constraint set $X$ into the objective function. In the following, we can derive the convergence of $dist(0, \partial P(x))$.

Let us denote its duality gap by $\widehat{\mathrm{Gap}}_k(x,y) = \hat{f}_k(x, \hat{y}_k(x)) - \hat{f}_k(\hat{x}_k(y), y)$, where we define $\hat{y}_k(x) := \arg\max_{y' \in Y} \hat{f}_k(x, y')$ given $x \in X$ and $\hat{x}_k(y) := \arg\min_{x' \in X} \hat{f}_k(x', y)$ given $y \in Y$. Its saddle point solution is denoted by $(\hat{x}_k^*, \hat{y}_k^*)$, i.e., $\hat{f}_k(\hat{x}_k^*, y) \leq \hat{f}_k(\hat{x}_k^*, \hat{y}_k^*) \leq \hat{f}_k(x, \hat{y}_k^*)$ for any $x \in X$ and $y \in Y$. The key idea of our analysis is to connect the duality gap $\widehat{\mathrm{Gap}}_k(x_0^k, y_0^k)$ to $\gamma^2\|\hat{x}_k^* - x_0^k\|^2$, and then by making $\gamma^2\|\hat{x}_k^* - x_0^k\|^2 \leq \epsilon^2$, we can show that $x_0^k$ is a nearly $\epsilon$-stationary point. To this end we first establish a bound of the duality gap for the regularized problem $\hat{f}_k(x, y)$ for the $k$-th epoch (Lemma 4). Then we connect it to $\gamma\|\hat{x}_k^* - x_0^k\|^2$ (Lemma 5). Finally, we bound $\gamma\|\hat{x}_k^* - x_0^k\|^2$ by a telescoping sum of $\mathrm{E}[\widehat{\mathrm{Gap}}_k(x_0^k, y_0^k)] - \mathrm{E}[\widehat{\mathrm{Gap}}_{k+1}(x_0^{k+1}, y_0^{k+1})]$ and $\mathrm{E}[P(x_0^k) - P(x_0^{k+1})]$.

**Lemma 4.** *Suppose Assumption 3 holds and $\gamma = 2\rho$. For $k \geq 1$, Lines 4 to 8 of Algorithm 2 guarantee*

$$\mathrm{E}[\widehat{Gap}_k(\bar{x}_k, \bar{y}_k)] = \mathrm{E}[\max_{y \in Y} \hat{f}_k(\bar{x}_k, y) - \min_{x \in X} \hat{f}_k(x, \bar{y}_k)] = \mathrm{E}[\hat{f}_k(\bar{x}_k, \hat{y}_k(\bar{x}_k)) - \hat{f}_k(\hat{x}_k(\bar{y}_k), \bar{y}_k)]$$

$$\leq \frac{5\eta_x^k M_1^2}{2} + \frac{5\eta_y^k M_2^2}{2} + \frac{1}{T_k}\left\{ (\frac{1}{\eta_x^k} + \frac{\rho}{2})\mathrm{E}[\|\hat{x}_k(\bar{y}_k) - x_0^k\|^2] + \frac{1}{\eta_y^k}\mathrm{E}[\|\hat{y}_k(\bar{x}_k) - y_0^k\|^2] \right\}. \quad (5)$$

For RHS of (5), particularly, due to $k \geq 1$, $T_k = \frac{106(k+1)}{3}$, $\eta_x^k = \frac{4}{\rho(k+1)}$ and $\eta_y^k = \frac{2}{\lambda(k+1)}$ in Algorithm 2, we have $\frac{1}{T_k}(\frac{1}{\eta_x^k} + \frac{\rho}{2}) \leq \frac{3\rho}{212}$ and $\frac{1}{T_k \eta_y^k} = \frac{3\lambda}{212}$. Then for the last two terms in the RHS of (5), we could have the following upper bound by the key lemma (Lemma 1)

$$\frac{3}{53}\left( \frac{\rho}{4}\|\hat{x}_k(\bar{y}_k) - x_0^k\|^2 + \frac{\lambda}{4}\|\hat{y}_k(\bar{x}_k) - y_0^k\|^2 \right) \leq \frac{3}{53}\left( \widehat{\mathrm{Gap}}_k(x_0^k, y_0^k) + \widehat{\mathrm{Gap}}_k(\bar{x}_k, \bar{y}_k) \right). \quad (6)$$

On the other hand, the following lemma lower bounds LHS of (5) to construct telescoping sums.

**Lemma 5.** *We could derive the following lower bound for $\widehat{Gap}_k(\bar{x}_k, \bar{y}_k)$*

$$\widehat{Gap}_k(\bar{x}_k, \bar{y}_k) \geq \frac{3}{50}\widehat{Gap}_{k+1}(x_0^{k+1}, y_0^{k+1}) + \frac{4}{5}(P(x_0^{k+1}) - P(x_0^k)) + \frac{\gamma}{80}\|x_0^k - \hat{x}_k^*\|^2. \quad (7)$$

Lemma 5 lower bounds $\widehat{\mathrm{Gap}}_k(\bar{x}_k, \bar{y}_k)$ in LHS of (5) by three parts. The first part constructs telescoping sum of $\widehat{\mathrm{Gap}}_{k+1}(x_0^{k+1}, y_0^{k+1}) - \widehat{\mathrm{Gap}}_k(x_0^k, y_0^k)$ together with (6). The second part itself is an element of telescoping sums over the primal gap. The third part $\|x_0^k - \hat{x}_k^*\|^2$ can be used as the measure of nearly $\epsilon$-stationary point, which is further explored in Theorem 2.

*Proof.* (of Theorem 2) Consider the $k$-th stage. Let us start from (5) in Lemma 4 as follows

$$\mathrm{E}[\widehat{\mathrm{Gap}}_k(\bar{x}_k, \bar{y}_k)]$$

$$\leq \frac{5\eta_x^k M_1^2}{2} + \frac{5\eta_y^k M_2^2}{2} + \frac{1}{T_k}\left\{ (\frac{1}{\eta_x} + \frac{\rho}{2})\mathrm{E}[\|\hat{x}_k(\bar{y}_k) - x_0^k\|^2] + \frac{1}{\eta_y}\mathrm{E}[\|\hat{y}_k(\bar{x}_k) - y_0^k\|^2] \right\}$$

$$\overset{(a)}{\leq} \frac{5\eta_x^k M_1^2}{2} + \frac{5\eta_y^k M_2^2}{2} + \frac{3}{53}\left( \frac{\rho}{4}\mathrm{E}[\|\hat{x}_k(\bar{y}_k) - x_0^k\|^2] + \frac{\lambda}{4}\mathrm{E}[\|\hat{y}_k(\bar{x}_k) - y_0^k\|^2] \right)$$

$$\overset{(6)}{\leq} \frac{5\eta_x^k M_1^2}{2} + \frac{5\eta_y^k M_2^2}{2} + \frac{3}{53}\mathrm{E}[\widehat{\mathrm{Gap}}_k(x_0^k, y_0^k)] + \frac{3}{53}\mathrm{E}[\widehat{\mathrm{Gap}}_k(\bar{x}_k, \bar{y}_k)],$$

where $(a)$ is due to settings $T_k = \frac{106(k+1)}{3}$, $\eta_x^k = \frac{4}{\rho(k+1)}$, and $\eta_y^k = \frac{2}{\lambda(k+1)}$. Re-organizing the above inequality, we have

$$\frac{50}{53}\mathrm{E}[\widehat{\mathrm{Gap}}_k(\bar{x}_k, \bar{y}_k)] \leq \frac{5\eta_x^k M_1^2}{2} + \frac{5\eta_y^k M_2^2}{2} + \frac{3}{53}\mathrm{E}[\widehat{\mathrm{Gap}}_k(x_0^k, y_0^k)]. \tag{8}$$

Then for the LHS of (8), we apply (7) of Lemma 5 as follows

$$\frac{50}{53}\Big(\frac{3}{50}\widehat{\mathrm{Gap}}_{k+1}(x_0^{k+1}, y_0^{k+1}) + \frac{4}{5}(P(x_0^{k+1}) - P(x_0^k)) + \frac{\gamma}{80}\|x_0^k - \hat{x}_k^*\|^2\Big)$$

$$\leq \frac{5\eta_x^k M_1^2}{2} + \frac{5\eta_y^k M_2^2}{2} + \frac{3}{53}\mathrm{E}[\widehat{\mathrm{Gap}}_k(x_0^k, y_0^k)]. \tag{9}$$

Next we have

$$\frac{5\gamma}{424}\mathrm{E}[\|x_0^k - \hat{x}_k^*\|^2] \leq \frac{5\eta_x^k M_1^2}{2} + \frac{5\eta_y^k M_2^2}{2} + \frac{40}{53}\mathrm{E}[P(x_0^k) - P(x_0^{k+1})]$$

$$+ \frac{3}{53}\Big(\mathrm{E}[\widehat{\mathrm{Gap}}_k(x_0^k, y_0^k)] - \mathrm{E}[\widehat{\mathrm{Gap}}_{k+1}(x_0^{k+1}, y_0^{k+1})]\Big) \tag{10}$$

Summing from $k = 1$ to $k = K$, we have

$$\frac{5\gamma}{424}\sum_{k=1}^{K}\mathrm{E}[\|x_0^k - \hat{x}_k^*\|^2] \leq \underbrace{\sum_{k=1}^{K}\frac{5\eta_x^k M_1^2}{2} + \sum_{k=1}^{K}\frac{5\eta_y^k M_2^2}{2}}_{:=A} + \underbrace{\frac{40}{53}\sum_{k=1}^{K}\mathrm{E}[P(x_0^k) - P(x_0^{k+1})]}_{:=B}$$

$$+ \underbrace{\frac{3}{53}\sum_{k=1}^{K}\Big(\mathrm{E}[\widehat{\mathrm{Gap}}_k(x_0^k, y_0^k)] - \mathrm{E}[\widehat{\mathrm{Gap}}_{k+1}(x_0^{k+1}, y_0^{k+1})]\Big)}_{:=C} \tag{11}$$

$$\leq 5\Big(\frac{2M_1^2}{\rho} + \frac{M_2^2}{\lambda}\Big)\ln(K+1) + \frac{43}{53}\mathrm{E}[\mathrm{Gap}(x_0^1, y_0^1)], \tag{12}$$

where the last inequality is due to the upper bounds the three terms $A$, $B$ and $C$ as follows.

For the term $A$, we have

$$A = \sum_{k=1}^{K}\frac{5\eta_x^k M_1^2}{2} + \sum_{k=1}^{K}\frac{5\eta_y^k M_2^2}{2} = \frac{10M_1^2}{\rho}\sum_{k=1}^{K}\frac{1}{k+1} + \frac{5M_2^2}{\lambda}\sum_{k=1}^{K}\frac{1}{k+1}$$

$$\leq 5\Big(\frac{2M_1^2}{\rho} + \frac{M_2^2}{\lambda}\Big)\ln(K+1),$$

where the second equality is due to the setting of $\eta_x^k = \frac{4}{\rho(k+1)}$ and $\eta_y^k = \frac{2}{\lambda(k+1)}$. The last inequality is due to $\sum_{k=1}^{K+1}\frac{1}{k} \leq \ln(K+1) + 1$.

For the term $B$, we have

$$B = \sum_{k=1}^{K}\mathrm{E}[P(x_0^k) - P(x_0^{k+1})] = \mathrm{E}[P(x_0^1) - P(x_0^{K+1})] = \mathrm{E}[f(x_0^1, \hat{y}(x_0^1)) - f(x_0^{K+1}, \hat{y}(x_0^{K+1}))]$$

$$\leq \mathrm{E}[f(x_0^1, \hat{y}(x_0^1)) - f(x_0^{K+1}, y_0^1)] \leq \mathrm{E}[f(x_0^1, \hat{y}(x_0^1)) - f(\hat{x}(y_0^1), y_0^1)] = \mathrm{E}[\mathrm{Gap}(x_0^1, y_0^1)],$$

where the two inequalities are due to $f(x_0^{K+1}, \hat{y}(x_0^{K+1})) \geq f(x_0^{K+1}, y_0^1) \geq f(\hat{x}(y_0^1), y_0^1)$.

For the term $C$, we have

$$C = \sum_{k=1}^{K}\Big(\mathrm{E}[\widehat{\mathrm{Gap}}_k(x_0^k, y_0^k) - \widehat{\mathrm{Gap}}_{k+1}(x_0^{k+1}, y_0^{k+1})]\Big)$$

$$= \mathrm{E}[\widehat{\mathrm{Gap}}_k(x_0^1, y_0^1) - \widehat{\mathrm{Gap}}_{K+1}(x_0^{K+1}, y_0^{K+1})] \leq \mathrm{E}[\widehat{\mathrm{Gap}}_k(x_0^1, y_0^1)]$$

$$= \mathrm{E}[f(x_0^1, \hat{y}(x_0^1)) + \frac{\gamma}{2}\|x_0^1 - x_0^1\|^2 - f(\hat{x}_1(y_0^1), y_0^1) - \frac{\gamma}{2}\|\hat{x}_1(y_0^1) - x_0^1\|^2]$$

$$\leq \mathrm{E}[f(x_0^1, \hat{y}(x_0^1)) - f(\hat{x}(y_0^1), y_0^1)] = \mathrm{E}[\mathrm{Gap}(x_0^1, y_0^1)],$$

where the first inequality is due to $\widehat{\mathrm{Gap}}_{K+1}(x_0^{K+1}, y_0^{K+1}) \geq 0$. By plugging the above upper bounds of the three terms $A$, $B$ and $C$ into (11), we have (12).

Then by randomly sampling $\tau$ from $\{1, ..., K\}$, we have

$$\mathrm{E}[\|x_0^\tau - \hat{x}_\tau^*\|^2] \leq \frac{424}{\gamma K}\Big(\frac{2M_1^2}{\rho} + \frac{M_2^2}{\lambda}\Big) \ln(K+1) + \frac{344}{5\gamma K}\mathrm{E}[\mathrm{Gap}(x_0^1, y_0^1)].$$

Since $\mathrm{E}[\mathrm{Dist}(0, \partial P(\hat{x}_\tau^*))^2] \leq \gamma^2 \mathrm{E}[\|x_\tau^* - x_0^\tau\|^2]$ and $\gamma = 2\rho$, we could set

$$K = \max\left\{\frac{1696\rho(\frac{2M_1^2}{\rho} + \frac{M_2^2}{\lambda})}{\epsilon^2}\ln\Big(\frac{1696\rho(\frac{2M_1^2}{\rho} + \frac{M_2^2}{\lambda})}{\epsilon^2}\Big), \frac{1376\rho\mathrm{Gap}(x_0^1, y_0^1)}{5\epsilon^2}\right\},$$

which leads to $\gamma^2\mathrm{E}[\|x_\tau^* - x_0^\tau\|^2] \leq \epsilon^2$. Recall $T_k = \frac{106(k+1)}{3}$. To compute the total number of iterations, we have

$$T_{tot} = \sum_{k=1}^{K} T_k = \frac{106}{3}\sum_{k=1}^{K}(k+1) = O(K^2) = \tilde{O}\left(\frac{1}{\epsilon^4}\right).$$

We would highlight that we prove the expectation result for WCSC in Theorem 2 for consistency with previous results [36]. Theorem 2 can also be extended to the high probability statement, as Theorem 1. In particular, we can prove a high-probability result of Lemma 4 similar to Lemma 2. Then by appropriately setting the radius $R_k$ according to $\eta_k$ and $T_k$ we can prove a similar result as in Lemma 3, which leads to a high-probability upper bound for the duality gap of $\hat{f}_k(x, y)$. From this point, we can prove the high-prob convergence for the WCSC similar to the existing proof of Theorem 2 except replacing expectation result with high-probability result. We will leave the detailed proof to the longer version. $\square$

## B   Proof of Lemma 1

*Proof.* Let us first consider the first term in LHS of (2) as follows,

$$\frac{\mu}{4}\|\hat{x}(y_1) - x_0\|^2$$
$$\leq \frac{\mu}{2}\|\hat{x}(y_1) - x^*\|^2 + \frac{\mu}{2}\|x^* - x_0^k\|^2$$
$$\overset{(a)}{\leq} f(x^*, y_1) - f(\hat{x}(y_1), y_1) + f(x_0, y^*) - f(x^*, y^*)$$
$$\overset{(b)}{\leq} f(x^*, y^*) - f(\hat{x}(y_1), y_1) + f(x_0, y^*) - f(x^*, y^*)$$
$$\overset{(c)}{\leq} f(x_0, \hat{y}(x_0)) - f(\hat{x}(y_1), y_1), \tag{13}$$

where inequality $(a)$ is due to $\mu$-strong convexity of $f(x, y_1)$ in $x$ with fixed $y_1$ (with optimality at $\hat{x}(y_1)$) and $\mu$-strong convexity of $f(x, y^*)$ in $x$ with fixed $y^*$ (with optimality at $x^*$). Inequality $(b)$ is due to $f(x^*, y_1) \leq f(x^*, y^*)$. Inequality $(c)$ is due to $f(x_0, y^*) \leq f(x_0, \hat{y}(x_0))$.

In a similar way, for the second term, we have

$$\frac{\lambda}{4}\|\hat{y}(x_1) - y_0\|^2$$
$$\leq \frac{\lambda}{2}\|\hat{y}(x_1) - y^*\|^2 + \frac{\lambda}{2}\|y^* - y_0\|^2$$
$$\overset{(a)}{\leq} f(x_1, \hat{y}(x_1)) - f(x_1, y^*) + f(x^*, y^*) - f(x^*, y_0)$$
$$\overset{(b)}{\leq} f(x_1, \hat{y}(x_1)) - f(x^*, y^*) + f(x^*, y^*) - f(x^*, y_0)$$
$$\overset{(c)}{\leq} f(x_1, \hat{y}(x_1)) - f(\hat{x}(y_0), y_0), \tag{14}$$

where inequality $(a)$ is due to $\lambda$-strong concavity of $f(x_1, y)$ in $y$ with fixed $x_1$ (optimality at $\hat{y}(x_1)$) and $f(x^*, y)$ in $y$ with fixed $x^*$ (optimality at $\hat{y}^*$). Inequality $(b)$ is due to $f(x_1, y^*) \geq f(x^*, y^*)$. Inequality $(c)$ is due to $f(x^*, y_0) \geq f(\hat{x}(y_0), y_0)$.

Then, combining inequalities (13) and (14), we have

$$\frac{\mu}{4}\|\hat{x}(y_1) - x_0\|^2 + \frac{\lambda}{4}\|\hat{y}(x_1) - y_0\|^2$$

$$\leq f(x_0, \hat{y}(x_0)) - f(\hat{x}(y_1), y_1) + f(x_1, \hat{y}(x_1)) - f(\hat{x}(y_0), y_0)$$

$$= \left( \max_{y' \in \Omega_2} f(x_0, y') - \min_{x' \in \Omega_1} f(x', y_0) \right) + \left( \max_{y' \in \Omega_2} f(x_1, y') - \min_{x' \in \Omega_1} f(x', y_1) \right).$$

□

## C  Proof of Lemma 2

*Proof.* Before the proof, we first present the following two lemmas as follows.

**Lemma 6.** *Let* $X_1, X_2, ..., X_T$ *be independent random variables and* $\mathrm{E}_t[\exp(\frac{X_t^2}{B^2})] \leq \exp(1)$ *for any* $t \in \{1, ..., T\}$. *Then we have with probability at least* $1 - \tilde{\delta}$

$$\sum_{t=1}^{T} X_t \leq B^2(T + \log(1/\tilde{\delta})).$$

**Lemma 7.** *(Lemma 2 of [24]) Let* $X_1, ..., X_T$ *be a martingale difference sequence, i.e.,* $E_t[X_t] = 0$ *for all t. Suppose that for some values* $\sigma_t$, *for* $t = 1, 2, ..., T$, *we have* $E_t[\exp(\frac{X_t^2}{\sigma_t^2})] \leq \exp(1)$. *Then with probability at least* $1 - \delta$, *we have*

$$\sum_{t=1}^{T} X_t \leq \sqrt{3\log(1/\delta) \sum_{t=1}^{T} \sigma_t^2}.$$

For simplicity of presentation, we use the notations $\Delta_x^t = \partial_x f(x_t, y_t; \xi_t)$, $\Delta_y^t = \partial y f(x_t, y_t, ; \xi_t)$, $\partial_x^t = \partial_x f(x_t, y_t)$ and $\partial_y^t = \partial_y f(x_t, y_t)$. To prove Lemma 2, we would leverage the following two update approaches:

$$\begin{cases} x_{t+1} = \arg\min_{x \in X \cap \mathcal{B}(x_0, R)} & x^\top \Delta_x^t + \frac{1}{2\eta_x}\|x - x_t\|^2 \\ y_{t+1} = \arg\min_{y \in Y \cap \mathcal{B}(y_0, R)} & -y^\top \Delta_y^t + \frac{1}{2\eta_y}\|y - y_t\|^2 \end{cases}$$

$$\begin{cases} \tilde{x}_{t+1} = \arg\min_{x \in X \cap \mathcal{B}(x_0, R)} & x^\top (\partial_x^t - \Delta_x^t) + \frac{1}{2\eta_x}\|x - \tilde{x}_t\|^2 \\ \tilde{y}_{t+1} = \arg\min_{y \in Y \cap \mathcal{B}(y_0, R)} & -y^\top (\partial_y^t - \Delta_y^t) + \frac{1}{2\eta_y}\|y - \tilde{y}_t\|^2, \end{cases} \quad (15)$$

where $x_0 = \tilde{x}_0$ and $y_0 = \tilde{y}_0$. The first two updates are identical to Line 4 and Line 5 in Algorithm 1. This can be verified easily. Take the first one as example:

$$x_{t+1} = \Pi_X(x_t - \eta_x \Delta_x^t) = \arg\min_{x \in X \cap \mathcal{B}(x_0, R)} \|x - (x_t - \eta_x \Delta_x^t)\|^2$$

$$= \arg\min_{x \in X \cap \mathcal{B}(x_0, R)} \frac{1}{2\eta_x}\|x - x_t\|^2 + x^\top \Delta_x^t.$$

Let $\psi(x) = x^\top u + \frac{1}{2\gamma}\|x - v\|^2$ with $x' = \arg\min_{x \in X'} \psi(x)$, which includes the four update approaches in (15) as special cases. By using the strong convexity of $\psi(x)$ and the first order optimality condition $(\partial\psi(x')^\top(x - x') \geq 0)$, for any $x \in X'$, we have

$$\psi(x) - \psi(x') \geq \partial\psi(x')^T(x - x') + \frac{1}{2\gamma}\|x - x'\|^2 \geq \frac{1}{2\gamma}\|x - x'\|^2,$$

which implies

$$
\begin{aligned}
0 \leq & (x - x')^\top u + \frac{1}{2\gamma}||x - v||^2 - \frac{1}{2\gamma}||x' - v||^2 - \frac{1}{2\gamma}||x - x'||^2 \\
= & (v - x')^\top u - (v - x)^\top u + \frac{1}{2\gamma}||x - v||^2 - \frac{1}{2\gamma}||x' - v||^2 - \frac{1}{2\gamma}||x - x'||^2 \\
= & -\frac{1}{2\gamma}||x' - v||^2 + (v - x')^\top u + \frac{1}{2\gamma}||x - v||^2 - \frac{1}{2\gamma}||x - x'||^2 - (v - x)^\top u \\
\leq & \frac{\gamma}{2}||u||^2 + \frac{1}{2\gamma}||x - v||^2 - \frac{1}{2\gamma}||x - x'||^2 - (v - x)^\top u.
\end{aligned}
$$

Then

$$
(v - x)^\top u \leq \frac{\gamma}{2}||u||^2 + \frac{1}{2\gamma}||x - v||^2 - \frac{1}{2\gamma}||x - x'||^2. \tag{16}
$$

Applying the above result to the updates in (15), we have for any $x \in X \cap \mathcal{B}(x_0, R)$ and $y \in Y \cap \mathcal{B}(y_0, R)$,

$$
\begin{aligned}
(x_t - x)^\top \Delta_x^t &\leq \frac{1}{2\eta_x}||x - x_t||^2 - \frac{1}{2\eta_x}||x - x_{t+1}||^2 + \frac{\eta_x}{2}||\Delta_x^t||^2 \\
(y - y_t)^\top \Delta_y^t &\leq \frac{1}{2\eta_y}||y - y_t||^2 - \frac{1}{2\eta_y}||y - y_{t+1}||^2 + \frac{\eta_y}{2}||\Delta_y^t||^2 \\
(\tilde{x}_t - x)^\top (\partial_x^t - \Delta_x^t) &\leq \frac{1}{2\eta_x}||x - \tilde{x}_t||^2 - \frac{1}{2\eta_x}||x - \tilde{x}_{t+1}||^2 + \frac{\eta_x}{2}||\partial_x^t - \Delta_x^t||^2 \\
(y - \tilde{y}_t)^\top (\partial_y^t - \Delta_y^t) &\leq \frac{1}{2\eta_y}||y - \tilde{y}_t||^2 - \frac{1}{2\eta_y}||y - \tilde{y}_{t+1}||^2 + \frac{\eta_y}{2}||\partial_y^t - \Delta_y^t||^2. \tag{17}
\end{aligned}
$$

Adding the above four inequalities together, we have

$$
\begin{aligned}
\text{LHS} = & (x_t - x)^\top \Delta_x^t + (y - y_t)^\top \Delta_y^t + (\tilde{x}_t - x)^\top (\partial_x^t - \Delta_x^t) + (y - \tilde{y}_t)^\top (\partial_y^t - \Delta_y^t) \\
= & (x_t - x)^\top \partial_x^t + (x_t - x)^\top (\Delta_x^t - \partial_x^t) + (y - y_t)^\top \partial_y^t + (y - y_t)^\top (\Delta_y^t - \partial_y^t) \\
& + (\tilde{x}_t - x)^\top (\partial_x^t - \Delta_x^t) + (y - \tilde{y}_t)^\top (\partial_y^t - \Delta_y^t) \\
= & -(x - x_t)^\top \partial_x^t + (y - y_t)^\top \partial_y^t - (x_t - \tilde{x}_t)^\top (\partial_x^t - \Delta_x^t) - (\tilde{y}_t - y_t)^\top (\partial_y^t - \Delta_y^t) \\
\overset{(a)}{\geq} & -(f(x, y_t) - f(x_t, y_t)) + (f(x_t, y) - f(x_t, y_t)) - (x_t - \tilde{x}_t)^\top (\partial_x^t - \Delta_x^t) - (\tilde{y}_t - y_t)^\top (\partial_y^t - \Delta_y^t) \\
= & f(x_t, y) - f(x, y_t) - (x_t - \tilde{x}_t)^\top (\partial_x^t - \Delta_x^t) - (\tilde{y}_t - y_t)^\top (\partial_y^t - \Delta_y^t) \\
\text{RHS} = & \frac{1}{2\eta_x}\left\{||x - x_t||^2 - ||x - x_{t+1}||^2 + ||x - \tilde{x}_t||^2 - ||x - \tilde{x}_{t+1}||^2\right\} + \frac{\eta_x}{2}\left\{||\Delta_x^t||^2 + ||\partial_x^t - \Delta_x^t||^2\right\} \\
& + \frac{1}{2\eta_y}\left\{||y - y_t||^2 - ||y - y_{t+1}||^2 + ||y - \tilde{y}_t||^2 - ||y - \tilde{y}_{t+1}||^2\right\} + \frac{\eta_y}{2}\left\{||\Delta_y^t||^2 + ||\partial_y^t - \Delta_y^t||^2\right\} \\
\overset{(b)}{\leq} & \frac{1}{2\eta_x}\left\{||x - x_t||^2 - ||x - x_{t+1}||^2 + ||x - \tilde{x}_t||^2 - ||x - \tilde{x}_{t+1}||^2\right\} + \frac{\eta_x}{2}\left\{3||\Delta_x^t||^2 + 2||\partial_x^t||^2\right\} \\
& + \frac{1}{2\eta_y}\left\{||y - y_t||^2 - ||y - y_{t+1}||^2 + ||y - \tilde{y}_t||^2 - ||y - \tilde{y}_{t+1}||^2\right\} + \frac{\eta_y}{2}\left\{3||\Delta_y^t||^2 + 2||\partial_y^t||^2\right\}
\end{aligned}
$$
$$\tag{18}$$

where inequality $(a)$ above is due to the convexity of $f(x, y_t)$ in $x$ and concavity of $f(x_t, y)$ in $y$. Inequality $(b)$ is due to $(a + b)^2 \leq 2a^2 + 2b^2$.

Then we combine the LHS and RHS by summing up $t = 0, ..., T - 1$:

$$\sum_{t=0}^{T-1}(f(x_t, y) - f(x, y_t)) \le \frac{1}{2\eta_x}\Big\{||x - x_0||^2 - ||x - x_T||^2 + ||x - \tilde{x}_0||^2 - ||x - \tilde{x}_T||^2\Big\}$$

$$\frac{1}{2\eta_y}\Big\{||y - y_0||^2 - ||y - y_T||^2 + ||y - \tilde{y}_0||^2 - ||y - \tilde{y}_T||^2\Big\}$$

$$+ \frac{3\eta_x}{2}\underbrace{\sum_{t=1}^{T}||\Delta_x^t||^2}_{:=A} + \eta_x\underbrace{\sum_{t=1}^{T}||\partial_x^t||^2}_{:=B}$$

$$+ \frac{3\eta_y}{2}\underbrace{\sum_{t=1}^{T}||\Delta_y^t||^2}_{:=C} + \eta_y\underbrace{\sum_{t=1}^{T}||\partial_y^t||^2}_{:=D}$$

$$+ \underbrace{\sum_{t=0}^{T-1}\Big((x_t - \tilde{x}_t)^\top(\partial_x^t - \Delta_x^t) + (y_t - \tilde{y}_t)^\top(\partial_y^t - \Delta_y^t)\Big)}_{:=E}. \qquad (19)$$

In the following, we show how to bound the above $A$ to $E$ terms. To bound the above term $A$ in (19), we apply Lemma 6 as follows, which holds with probability $1 - \tilde{\delta}$,

$$\sum_{t=1}^{T}||\Delta_x^t||^2 \le B_1^2(T + \log(1/\tilde{\delta})). \qquad (20)$$

Similarly, term $C$ in (19) can be bounded with probability $1 - \tilde{\delta}$ as follows

$$\sum_{t=1}^{T}||\Delta_y^t||^2 \le B_2^2(T + \log(1/\tilde{\delta})). \qquad (21)$$

To bound term $B$ of (19), which contains only the full subgradients $\partial_x^t$, we have

$$||\partial_x^t||^2 = ||\mathrm{E}[\Delta_x^t]||^2 \le \mathrm{E}[||\Delta_x^t||^2] \le B_1^2,$$

where the first inequality is due to Jensen's inequality and the second inequality is due to

$$\exp(\mathrm{E}[\frac{||\Delta_x^t||^2}{B_1^2}]) \le \mathrm{E}[\exp(\frac{||\Delta_x^t||^2}{B_1^2})] \le \exp(1) \;\; \Rightarrow \;\; \mathrm{E}[\frac{||\Delta_x^t||^2}{B_1^2}] \le 1 \;\; \Rightarrow \;\; \mathrm{E}[||\Delta_x^t||^2] \le B_1^2.$$

Therefore, we have

$$\sum_{t=1}^{T}||\partial_x^t||^2 \le TB_1^2. \qquad (22)$$

Similarly, for term $D$ in (19), we have

$$\sum_{t=1}^{T}||\partial_y^t||^2 \le TB_2^2. \qquad (23)$$

To bound term $E$ of (19), let $U_t = (x_t - \tilde{x}_t)^\top(\partial_x^t - \Delta_x^t)$ and $V_t = (y_t - \tilde{y}_t)^\top(\partial_y^t - \Delta_y^t)$ for $t \in \{0, ..., T - 1\}$, which are Martingale difference sequences. We thus would like to use Lemma 7 to handle these terms. To this end, we can first upper bound $|U_t|$ and $|V_t|$ as follows

$$|U_t| = |(x_t - \tilde{x}_t)^\top(\partial_x^t - \Delta_x^t)| \le ||x_t - x_0 + x_0 - \tilde{x}_t|| \cdot ||\partial_x^t - \Delta_x^t||$$

$$\le 2R(||\partial_x^t|| + ||\Delta_x^t||) \le 2R(B_1 + ||\Delta_x^t||),$$

$$|V_t| = |(y_t - \tilde{y}_t)^\top(\partial_y^t - \Delta_y^t)| \le ||(y_t - y_0 + y_0 - \tilde{y}_t|| \cdot ||\partial_y^t - \Delta_y^t)||$$

$$\le 2R(||\partial_y^t|| + ||\Delta_y^t)||) \le 2R(B_2 + ||\Delta_y^t)||).$$

Then the above two inequalities implies that

$$\mathrm{E}_t[\exp(\frac{U_t^2}{16B_1^2R^2})] \leq \mathrm{E}_t[\exp(\frac{(2R(B_1+\|\Delta_x^t\|))^2}{16B_1^2R^2})] \overset{(a)}{\leq} \mathrm{E}_t[\exp(\frac{4R^2(2B_1^2+2\|\Delta_x^t\|^2)}{16B_1^2R^2})]$$

$$=\mathrm{E}_t[\exp(\frac{B_1^2+\|\Delta_x^t\|^2}{2B_1^2})] = \mathrm{E}_t[\exp(\frac{1}{2}+\frac{\|\Delta_x^t\|^2}{2B_1^2})]$$

$$=\exp(\frac{1}{2}) \cdot \mathrm{E}_t\Big[\sqrt{\exp(\frac{\|\Delta_x^t\|^2}{B_1^2})}\Big] \overset{(b)}{\leq} \exp(\frac{1}{2}) \cdot \sqrt{\mathrm{E}_t[\exp(\frac{\|\Delta_x^t\|^2}{B_1^2})]}$$

$$\overset{(c)}{\leq} \exp(\frac{1}{2})\sqrt{\exp(1)} = \exp(1), \tag{24}$$

where inequality $(a)$ is due to $(a+b)^2 \leq 2a^2 + 2b^2$, inequality $(b)$ is due to the concavity of $\sqrt{\cdot}$ and Jensen's inequality. Inequality $(c)$ is due to the assumption. In a similar way, we have

$$\mathrm{E}_t[\exp(\frac{V_t^2}{16B_2^2R^2})] \leq \exp(1). \tag{25}$$

Next, applying Lemma 7 with (24) and (25), we have with probability at least $1-\tilde{\delta}$

$$\sum_{t=0}^{T-1} U_t \leq 4B_1 R\sqrt{3\log(1/\tilde{\delta})T},$$

$$\sum_{t=0}^{T-1} V_t \leq 4B_2 R\sqrt{3\log(1/\tilde{\delta})T}. \tag{26}$$

For LHS of (19), by Jensen's inequality, we have

$$\sum_{t=0}^{T-1}(f(x_t,y)-f(x,y_t)) \geq T(f(\bar{x},y)-f(x,\bar{y})), \tag{27}$$

where $\bar{x} = \frac{1}{T}\sum_{t=0}^{T-1}x_t$ and $\bar{y} = \frac{1}{T}\sum_{t=0}^{T-1}$.

Suppose $T \geq 1$. By plugging (27), (20), (21), (22), (23) and (26) back into (19), with probability at least $1-\tilde{\delta}$, we have

$$f(\bar{x},y)-f(x,\bar{y}) \leq \frac{\|x-x_0\|^2}{\eta_x T} + \frac{\|y-y_0\|^2}{\eta_y T} + \frac{\eta_x B_1^2}{2}(5+3\log(1/\tilde{\delta})) + \frac{\eta_y B_2^2}{2}(5+3\log(1/\tilde{\delta}))$$

$$+ \frac{4(B_1+B_2)R\sqrt{3\log(1/\tilde{\delta})}}{\sqrt{T}} \tag{28}$$

$\square$

# D   Proof of Lemma 6

*Proof.* First, we start from

$$\mathrm{E}[\exp(\frac{\sum_{t=1}^T X_t}{B^2})] = \mathrm{E}[\mathrm{E}_T[\exp(\frac{\sum_{t=1}^{T-1}X_t+X_T}{B^2})]]$$

$$=\mathrm{E}[\exp(\frac{\sum_{t=1}^{T-1}X_t}{B^2}) \cdot \mathrm{E}_T[\exp(\frac{X_T}{B^2})]]$$

$$\leq \mathrm{E}[\exp(\frac{\sum_{t=1}^{T-1}X_t}{B^2}) \cdot \exp(1)]$$

$$\leq \mathrm{E}[\exp(\frac{\sum_{t=1}^{T-2}X_t}{B^2}) \cdot \exp(2)]$$

$$\leq \exp(T),$$

where the first inequality is due to the assumption.

Markov inequality indicates that $P(X \geq a) \leq \frac{\mathrm{E}[X]}{a}$ for a random variable $X$, which, by additionally introducing $\tilde{\delta}$, leads to

$$P\left(\exp(\frac{\sum_{t=1}^{T} X_t}{B^2}) \geq \frac{\mathrm{E}[\exp(\frac{\sum_{t=1}^{T} X_t}{B^2})]}{\tilde{\delta}}\right) \leq \tilde{\delta}.$$

Therefore, with probability at least $1 - \tilde{\delta}$, we have

$$\exp(\frac{\sum_{t=1}^{T} X_t}{B^2}) \leq \frac{\mathrm{E}[\exp(\frac{\sum_{t=1}^{T} X_t}{B^2})]}{\tilde{\delta}} \leq \frac{\exp(T)}{\tilde{\delta}}$$

$$\Rightarrow \frac{\sum_{t=1}^{T} X_t}{B^2} \leq \log(\frac{\exp(T)}{\tilde{\delta}}) = \log(\exp(T)) + \log(1/\tilde{\delta}) = T + \log(1/\tilde{\delta})$$

$$\Rightarrow \sum_{t=1}^{T} X_t \leq B^2(T + \log(1/\tilde{\delta})).$$

$\square$

# E    Proof of Lemma 3

*Proof.* Here we consider the following problem

$$\min_{x \in X \cap \mathcal{B}(x_0, R)} \max_{y \in Y \cap \mathcal{B}(y_0, R)} f(x, y)$$

with two solutions $(x_0, y_0)$ and $(\bar{x}, \bar{y})$.

By (1) of Lemma 1, we have

$$\frac{\mu}{4}\|\hat{x}_R(\bar{y}) - x_0\|^2 + \frac{\lambda}{4}\|\hat{y}_R(\bar{x}) - y_0\|^2 \leq \underbrace{\max_{y' \in Y \cap \mathcal{B}(y_0, R)} f(x_0, y') - \min_{x \in X \cap \mathcal{B}(x_0, R)} f(x', y_0)}_{:=A}$$

$$+ \underbrace{\max_{y' \in Y \cap \mathcal{B}(y_0, R)} f(\bar{x}, y') - \min_{x \in X \cap \mathcal{B}(x_0, R)} f(x', \bar{y})}_{:=B}. \quad (29)$$

We can bound the above term $A$ as follows

$$\max_{y' \in Y \cap \mathcal{B}(y_0, R)} f(x_0, y') - \min_{x \in X \cap \mathcal{B}(x_0, R)} f(x', y_0)$$

$$\leq \max_{y' \in Y} f(x_0, y') - \min_{x \in X} f(x', y_0) \leq \frac{\min\{\mu, \lambda\}R^2}{8}, \quad (30)$$

where the last inequality is due to the setting of $R$.

Recall the definitions

$$\hat{x}_R(\bar{y}) = \arg\min_{x' \in x \cap \mathcal{B}(x_0, R)} f(x', \bar{y}), \qquad \hat{y}_R(\bar{x}) = \arg\max_{y' \in Y \cap \mathcal{B}(y_0, R)} f(\bar{x}, y').$$

To Bound term $B$ in (29), we apply Lemma 2 as follows

$$\max_{y' \in Y \cap \mathcal{B}(y_0, R)} f(\bar{x}, y') - \min_{x' \in \cap \mathcal{B}(x_0, R)} f(x, \bar{y})$$

$$\leq \frac{\|\hat{x}_R(\bar{y}) - x_0\|^2}{\eta_x T} + \frac{\|\hat{y}_R(\bar{x}) - y_0\|^2}{\eta_y T} + \frac{\eta_x B_1^2}{2}(5 + 3\log(1/\tilde{\delta})) + \frac{\eta_y B_2^2}{2}(5 + 3\log(1/\tilde{\delta}))$$

$$+ \frac{4(B_1 + B_2)R\sqrt{2\log(1/\tilde{\delta})}}{\sqrt{T}}$$

$$\leq \frac{R^2}{\eta_x T} + \frac{R^2}{\eta_y T} + \frac{\eta_x B_1^2}{2}(5 + 3\log(1/\tilde{\delta})) + \frac{\eta_y B_2^2}{2}(5 + 3\log(1/\tilde{\delta}))$$

$$+ \frac{4(B_1 + B_2)R\sqrt{2\log(1/\tilde{\delta})}}{\sqrt{T}} \leq \frac{\min\{\mu, \lambda\}R^2}{16}, \quad (31)$$

where the last inequality holds with probability at least $1 - \tilde{\delta}$ with the setting of $\eta_x$, $\eta_y$ and $T$ as follows

$$\eta_x = \frac{\min\{\mu, \lambda\} R^2}{40(5 + 3\log(1/\tilde{\delta})) B_1^2}, \eta_y = \frac{\min\{\mu, \lambda\} R^2}{40(5 + 3\log(1/\tilde{\delta})) B_2^2}$$

$$T \geq \frac{\max\left\{320^2(B_1 + B_1)^2 3\log(1/\tilde{\delta}), 3200(5 + 3\log(1/\tilde{\delta})) \max\{B_1^2, B_2^2\}\right\}}{\min\{\mu, \lambda\}^2 R^2}. \tag{32}$$

Finally, we use (30) and (31) to bound term $A$ and term $B$ in (29) as follows

$$\frac{\mu}{4}\|\hat{x}_R(\bar{y}) - x_0\|^2 + \frac{\lambda}{4}\|\hat{y}_R(\bar{x}) - y_0\|^2 \leq \frac{\min\{\mu, \lambda\} R^2}{8} + \frac{\min\{\mu, \lambda\} R^2}{16} = \frac{3\min\{\mu, \lambda\} R^2}{16}$$

$$< \frac{\min\{\mu, \lambda\} R^2}{4}.$$

It implies

$$\|\hat{x}_R(\bar{y}) - x_0\| < R,$$
$$\|\hat{y}_R(\bar{x}) - y_0\| < R,$$

which shows $\hat{x}_R(\bar{y})$ and $\hat{y}_R(\bar{y})$ are interior points of $\mathcal{B}(x_0, R)$ and $\mathcal{B}(y_0, R)$, respectively, so that $\hat{x}_R(\bar{y}) = \hat{x}(\bar{y})$ and $\hat{y}_R(\bar{x}) = \hat{y}(\bar{x})$.

$\square$

# F   Proof of Theorem 1

*Proof.* Let

$$T_1 = \frac{\max\left\{320^2(B_1 + B_2)^2 3\log(1/\tilde{\delta}), 3200(3\log(1/\tilde{\delta}) + 2) \max\{B_1^2, B_2^2\}\right\}}{\min\{\mu, \lambda\}^2 R_1^2},$$

where $\mathrm{Gap}(x_0, y_0) = \max_{y \in Y} f(x_0, y) - \min_{x \in X} f(x, y_0) \leq \epsilon_0$ and $R_1 \geq 2\sqrt{\frac{2\epsilon_0}{\min\{\mu, \lambda\}}}$.

Given $T_{k+1} = 2T_k$ in Algorithm 1 and $K = \lceil \log(\frac{\epsilon_0}{\epsilon}) \rceil$, the total number of iterations can be computed by

$$T_{tot} = \sum_{k=1}^{K} T_k = T_1 \sum_{k=1}^{K} 2^{k-1} = T_1(2^K - 1) \leq T_1 2^{\lceil \log(\frac{\epsilon_0}{\epsilon}) \rceil} \leq T_1 \frac{2\epsilon_0}{\epsilon}$$

$$= \frac{\max\left\{320^2(B_1 + B_2)^2 3\log(1/\tilde{\delta}), 3200(3\log(1/\tilde{\delta}) + 2) \max\{B_1^2, B_2^2\}\right\}}{\min\{\mu, \lambda\}^2 R_1^2} \cdot \frac{2\epsilon_0}{\epsilon}$$

$$\leq \frac{\max\left\{320^2(B_1 + B_2)^2 3\log(1/\tilde{\delta}), 3200(3\log(1/\tilde{\delta}) + 2) \max\{B_1^2, B_2^2\}\right\}}{8\min\{\mu, \lambda\}\epsilon_0} \cdot \frac{2\epsilon_0}{\epsilon}$$

$$= \frac{\max\left\{320^2(B_1 + B_2)^2 3\log(\frac{1}{\tilde{\delta}}), 3200(3\log(1/\tilde{\delta}) + 2) \max\{B_1^2, B_2^2\}\right\}}{4\min\{\mu, \lambda\}\epsilon}$$

$\square$

# G   Proof of Lemma 4

*Proof.* In this proof, we focus on the analysis of one inner loop and thus omit the index of $k$ for simpler presentation. Let $\Delta_x^t = \partial_x f(x_t, y_t; \xi_t)$, $\Delta_y^t = \partial_y f(x_t, y_t; \xi^t)$, $\partial_x^t = \partial_x f(x_t, y_t)$ and $\partial_y^t = \partial_y f(x_t, y_t)$. Denote $\hat{f}(x, y) = f(x, y) + \frac{\gamma}{2}\|x - x_0\|^2$.

Let $\psi_x^t(x) = x^\top \Delta_x^t + \frac{1}{2\eta_x}\|x - x_t\|^2 + \frac{\gamma}{2}\|x - x_0\|^2$ and $\psi_y^t(y) = -y^\top \Delta_y^t + \frac{1}{2\eta_y}\|y - y_t\|^2$. According to the update of $x_{t+1}$ and $y_{t+1}$, we have $x_{t+1} = \arg\min_{x \in X} \psi_x^t(x)$ and $y_{t+1} = \arg\max_{y \in Y} \psi_y^t(y)$. It is easy to verify that $\psi_x^t$ and $\psi_y^t$ are strongly convex in $x$ and $y$, respectively.

By $\left(\frac{1}{\eta_x} + \gamma\right)$-strong convexity of $\psi_x^t(x)$ and the optimality condition at $x_{t+1}$, we have

$$\left(\frac{1}{2\eta_x} + \frac{\gamma}{2}\right)\|x - x_{t+1}\|^2 \le \psi_x^t(x) - \psi_x^t(x_{t+1})$$

$$= x^\top \Delta_x^t + \frac{1}{2\eta_x}\|x - x_t\|^2 + \frac{\gamma}{2}\|x - x_0\|^2 - \left(x_{t+1}^\top \Delta_x^t + \frac{1}{2\eta_x}\|x_{t+1} - x_t\|^2 + \frac{\gamma}{2}\|x_{t+1} - x_0\|^2\right)$$

$$= (x - x_t)^\top \partial_x^t + (x_t - x_{t+1})^\top \partial_x^t + (x - x_{t+1})^\top (\Delta_x^t - \partial_x^t)$$

$$+ \frac{1}{2\eta_x}\|x - x_t\|^2 + \frac{\gamma}{2}\|x - x_0\|^2 - \frac{1}{2\eta_x}\|x_{t+1} - x_t\|^2 - \frac{\gamma}{2}\|x_{t+1} - x_0\|^2$$

$$\overset{(a)}{\le} f(x, y_t) - f(x_t, y_t) + \frac{\gamma}{2}\|x - x_0\|^2 - \frac{\gamma}{2}\|x_t - x_0\|^2 + \frac{\gamma}{2}\left(\|x_t - x_0\|^2 - \|x_{t+1} - x_0\|^2\right)$$

$$+ \left(\frac{1}{2\eta_x} + \frac{\rho}{2}\right)\|x - x_t\|^2 + (x - x_t)^\top (\Delta_x^t - \partial^t) + (x_t - x_{t+1})^\top \Delta_x^t - \frac{1}{2\eta_x}\|x_{t+1} - x_t\|^2$$

$$\overset{(b)}{\le} \hat{f}(x, y_t) - \hat{f}(x_t, y_t) + \frac{\gamma}{2}\left(\|x_t - x_0\|^2 - \|x_{t+1} - x_0\|^2\right)$$

$$+ \left(\frac{1}{2\eta_x} + \frac{\rho}{2}\right)\|x - x_t\|^2 + (x - x_t)^\top (\Delta_x^t - \partial^t) + \frac{\eta_x}{2}\|\Delta_x^t\|^2, \tag{33}$$

where inequality $(a)$ is due to $\rho$-weakly convexity of $f$ in $x$. Inequality $(b)$ is due to Young's inequality, i.e., $(x_t - x_{t+1})^\top \Delta_x^t - \frac{1}{2\eta_x}\|x_{t+1} - x_t\|^2 \le \frac{\eta_x}{2}\|\Delta_x^t\|^2$.

Similarly, due to the $\frac{1}{\eta_y}$-strong convexity of $\psi_y^t$ in $y$ and the optimality condition of $y_{t+1}$, we have

$$\frac{1}{2\eta_y}\|y - y_{t+1}\|^2 \le \psi_y^t(y) - \psi_y^t(y_{t+1})$$

$$= -y^\top \Delta_y^t + \frac{1}{2\eta_y}\|y - y_t\|^2 - \left(-y_{t+1}^\top \Delta_y^t + \frac{1}{2\eta_y}\|y_{t+1} - y_t\|^2\right)$$

$$= (y_t - y)^\top \partial_y^t + (y_{t+1} - y_t)^\top \partial_y^t + (y_{t+1} - y)^\top (\Delta_y^t - \partial_y^t)$$

$$+ \frac{1}{2\eta_y}\|y - y_t\|^2 - \frac{1}{2\eta_y}\|y_{t+1} - y_t\|^2$$

$$\overset{(a)}{\le} f(x_t, y_t) - f(x_t, y) + (y_{t+1} - y_t)^\top \Delta_y^t + (y_t - y)^\top (\Delta_y^t - \partial_y^t)$$

$$+ \frac{1}{2\eta_y}\|y - y_t\|^2 - \frac{1}{2\eta_y}\|y_{t+1} - y_t\|^2$$

$$\overset{(b)}{\le} \hat{f}(x_t, y_t) - \hat{f}(x_t, y) + (y_t - y)^\top (\Delta_y^t - \partial_y^t) + \frac{1}{2\eta_y}\|y - y_t\|^2 + \frac{\eta_y}{2}\|\Delta_y^t\|^2, \tag{34}$$

where inequality $(a)$ is due to concavity of $f$ in $y$. Inequality $(b)$ is due to Young's inequality, i.e., $(y_{t+1} - y_t)^\top \Delta_y^t - \frac{1}{2\eta_y}\|y_{t+1} - y_t\|^2 \le \frac{\eta_y}{2}\|\Delta_y^t\|^2$.

Combining (33) and (34), we have

$$\hat{f}(x_t, y) - \hat{f}(x, y_t) \le \frac{\eta_x}{2}\|\Delta_x^t\|^2 + \frac{\eta_y}{2}\|\Delta_y^t\|^2$$

$$+ (x - x_t)^\top (\Delta_x^t - \partial^t) + (y_t - y)^\top (\Delta_y^t - \partial^t) + \frac{\gamma}{2}\left(\|x_t - x_0\|^2 - \|x_{t+1} - x_0\|^2\right)$$

$$+ \left(\frac{1}{2\eta_x} + \frac{\rho}{2}\right)\|x - x_t\|^2 - \left(\frac{1}{2\eta_x} + \frac{\gamma}{2}\right)\|x - x_{t+1}\|^2 + \frac{1}{2\eta_y}\left(\|y - y_t\|^2 - \|y - y_{t+1}\|^2\right). \tag{35}$$

Now we do not take expectation, since we aim to eliminate the randomness of $x$ and $y$ in $(x - x_t)$ and $(y_t - y)$, respectively. To achieve this, we use the following updates

$$\tilde{x}_{t+1} = \arg\min_{x \in X} x^\top (\partial_x^t - \Delta_x^t) + \frac{1}{2\eta_x}\|x - \tilde{x}_t\|^2$$

$$\tilde{y}_{t+1} = \arg\min_{y \in Y} -y^\top (\partial_y^t - \Delta_y^t) + \frac{1}{2\eta_y}\|y - \tilde{y}_t\|^2,$$

where $\tilde{x}_0 = x_0$ and $\tilde{y}_0 = y_0$.

Using similar analysis as the beginning, we have

$$\frac{1}{2\eta_x}\|x - \tilde{x}_{t+1}\|^2 \le x^\top(\partial_x^t - \Delta_x^t) + \frac{1}{2\eta_x}\|x - \tilde{x}_t\|^2 - \left(\tilde{x}_{t+1}^\top(\partial_x^t - \Delta_x^t) + \frac{1}{2\eta_x}\|\tilde{x}_{t+1} - \tilde{x}_t\|^2\right)$$

$$= (\tilde{x}_t - x)^\top(\Delta_x^t - \partial_x^t) + \frac{1}{2\eta_x}\|x - \tilde{x}_t\|^2 + (\tilde{x}_t - \tilde{x}_{t+1})^\top(\partial_x^t - \Delta_x^t) - \frac{1}{2\eta_x}\|\tilde{x}_{t+1} - \tilde{x}_t\|^2$$

$$\le (\tilde{x}_t - x)^\top(\Delta_x^t - \partial_x^t) + \frac{1}{2\eta_x}\|x - \tilde{x}_t\|^2 + \frac{\eta_x}{2}\|\partial_x^t - \Delta_x^t\|^2$$

$$\le (\tilde{x}_t - x)^\top(\Delta_x^t - \partial_x^t) + \frac{1}{2\eta_x}\|x - \tilde{x}_t\|^2 + \eta_x\|\partial_x^t\|^2 + \eta_x\|\Delta_x^t\|^2.$$

We could also derive the similar result for $y$ as follows

$$\frac{1}{2\eta_y}\|y - \tilde{y}_{t+1}\|^2 \le -y^\top(\partial_y^t - \Delta_y^t) + \frac{1}{2\eta_y}\|y - \tilde{y}_t\|^2 - \left(-\tilde{y}_{t+1}^\top(\partial_y^t - \Delta_y^t) + \frac{1}{2\eta_y}\|\tilde{y}_{t+1} - \tilde{y}_t\|^2\right)$$

$$= (y - \tilde{y}_t)^\top(\Delta_y^t - \partial_y^t) + \frac{1}{2\eta_y}\|y - \tilde{y}_t\|^2 + (\tilde{y}_{t+1} - \tilde{y}_t)^\top(\partial_y^t - \Delta_y^t) - \frac{1}{2\eta_y}\|\tilde{y}_{t+1} - \tilde{y}_t\|^2$$

$$\le (y - \tilde{y}_t)^\top(\Delta_y^t - \partial_y^t) + \frac{1}{2\eta_y}\|y - \tilde{y}_t\|^2 + \frac{\eta_y}{2}\|\partial_y^t - \Delta_y^t\|^2$$

$$\le (y - \tilde{y}_t)^\top(\Delta_y^t - \partial_y^t) + \frac{1}{2\eta_y}\|y - \tilde{y}_t\|^2 + \eta_y\|\partial_y^t\|^2 + \eta_y\|\Delta_y^t\|^2.$$

Summing the above two inequalities, we have

$$0 \le \frac{1}{2\eta_x}\left(\|x - \tilde{x}_t\|^2 - \|x - \tilde{x}_{t+1}\|^2\right) + (\tilde{x}_t - x)^\top(\Delta_x^t - \partial_x^t) + \eta_x\|\partial_x^t\|^2 + \eta_x\|\Delta_x^t\|^2$$

$$+ \frac{1}{2\eta_y}\left(\|y - \tilde{y}_t\|^2 - \|y - \tilde{y}_{t+1}\|^2\right) + (y - \tilde{y}_t)^\top(\Delta_y^t - \partial_y^t) + \eta_y\|\partial_y^t\|^2 + \eta_y\|\Delta_y^t\|^2 \quad (36)$$

Combining (35) and (36), we have

$$\hat{f}(x_t, y) - \hat{f}(x, y_t) \le \frac{\eta_x}{2}\|\Delta_x^t\|^2 + \frac{\eta_y}{2}\|\Delta_y^t\|^2$$

$$+ (x - x_t)^\top(\Delta_x^t - \partial_x^t) + (y_t - y)^\top(\Delta_y^t - \partial_y^t) + \frac{\gamma}{2}\left(\|x_t - x_0\|^2 - \|x_{t+1} - x_0\|^2\right)$$

$$+ \left(\frac{1}{2\eta_x} + \frac{\rho}{2}\right)\|x - x_t\|^2 - \left(\frac{1}{2\eta_x} + \frac{\gamma}{2}\right)\|x - x_{t+1}\|^2 + \frac{1}{2\eta_y}\left(\|y - y_t\|^2 - \|y - y_{t+1}\|^2\right)$$

$$+ \frac{1}{2\eta_x}\left(\|x - \tilde{x}_t\|^2 - \|x - \tilde{x}_{t+1}\|^2\right) + (\tilde{x}_t - x)^\top(\Delta_x^t - \partial_x^t) + \eta_x\|\partial_x^t\|^2 + \eta_x\|\Delta_x^t\|^2$$

$$+ \frac{1}{2\eta_y}\left(\|y - \tilde{y}_t\|^2 - \|y - \tilde{y}_{t+1}\|^2\right) + (y - \tilde{y}_t)^\top(\Delta_y^t - \partial_y^t) + \eta_y\|\partial_y^t\|^2 + \eta_y\|\Delta_y^t\|^2$$

$$= \frac{3\eta_x}{2}\|\Delta_x^t\|^2 + \eta_x\|\partial_x^t\|^2 + \frac{3\eta_y}{2}\|\Delta_y^t\|^2 + \eta_y\|\partial_y^t\|^2$$

$$+ (\tilde{x}_t - x_t)^\top(\Delta_x^t - \partial_x^t) + (y_t - \tilde{y}_t)^\top(\Delta_y^t - \partial_y^t) + \frac{\gamma}{2}\left(\|x_t - x_0\|^2 - \|x_{t+1} - x_0\|^2\right)$$

$$+ \left(\frac{1}{2\eta_x} + \frac{\rho}{2}\right)\|x - x_t\|^2 - \left(\frac{1}{2\eta_x} + \frac{\gamma}{2}\right)\|x - x_{t+1}\|^2 + \frac{1}{2\eta_y}\left(\|y - y_t\|^2 - \|y - y_{t+1}\|^2\right)$$

$$+ \frac{1}{2\eta_x}\left(\|x - \tilde{x}_t\|^2 - \|x - \tilde{x}_{t+1}\|^2\right) + \frac{1}{2\eta_y}\left(\|y - \tilde{y}_t\|^2 - \|y - \tilde{y}_{t+1}\|^2\right)$$

Summing the above inequality from $t = 0$ to $T - 1$ and using Jensen's inequality, we have

$$T\Big(\hat{f}(\bar{x}, y) - \hat{f}(x, \bar{y})\Big) \leq \sum_{t=0}^{T-1} \Big(\hat{f}(x_t, y) - \hat{f}(x, y_t)\Big)$$

$$\leq \frac{\eta_x}{2} \sum_{t=0}^{T-1} (3\|\Delta_x^t\|^2 + 2\|\partial_x^t\|^2) + \frac{\eta_y}{2} \sum_{t=0}^{T-1} (3\|\Delta_y^t\|^2 + 2\|\partial_y^t\|^2)$$

$$+ \sum_{t=0}^{T-1} (\tilde{x}_t - x_t)^\top (\Delta_x^t - \partial_x^t) + \sum_{t=0}^{T-1} (y_t - \tilde{y}_t)^\top (\Delta_y^t - \partial_y^t) + \frac{\gamma}{2}\Big(\|x_0 - x_0\|^2 - \|x_T - x_0\|^2\Big)$$

$$+ \Big(\frac{1}{2\eta_x} + \frac{\rho}{2}\Big)\|x - x_0\|^2 - \Big(\frac{1}{2\eta_x} + \frac{\gamma}{2}\Big)\|x - x_T\|^2 + \frac{1}{2\eta_y}\Big(\|y - y_0\|^2 - \|y - y_T\|^2\Big)$$

$$+ \frac{1}{2\eta_x}\Big(\|x - x_0\|^2 - \|x - \tilde{x}_T\|^2\Big) + \frac{1}{2\eta_y}\Big(\|y - y_0\|^2 - \|y - \tilde{y}_T\|^2\Big)$$

where $\bar{x} = \frac{1}{T}\sum_{t=0}^{T-1} x_t$ and $\bar{y} = \frac{1}{T}\sum_{t=0}^{T-1} y_t$.

Plugging in $x = \hat{x}(\bar{y})$ and $y = \hat{y}(\bar{x})$, we have

$$\widehat{\mathrm{Gap}}(\bar{x}, \bar{y}) = \hat{f}(\bar{x}, \hat{y}(\bar{x})) - \hat{f}(\hat{x}(\bar{y}), \bar{y}) \leq \frac{1}{T} \sum_{t=0}^{T-1} \Big(\hat{f}(x_t, \hat{y}(\bar{x})) - \hat{f}(\hat{x}(\bar{y}), y_t)\Big)$$

$$\leq \frac{\eta_x}{2T} \sum_{t=0}^{T-1} (3\|\Delta_x^t\|^2 + 2\|\partial_x^t\|^2) + \frac{\eta_y}{2T} \sum_{t=0}^{T-1} (3\|\Delta_y^t\|^2 + 2\|\partial_y^t\|^2)$$

$$+ \frac{1}{T} \sum_{t=0}^{T} (\tilde{x}_t - x_t)^\top (\Delta_x^t - \partial_x^t) + \frac{1}{T} \sum_{t=0}^{T} (y_t - \tilde{y}_t)^\top (\Delta_y^t - \partial_y^t)$$

$$+ \frac{1}{T}\Big(\frac{1}{\eta_x} + \frac{\rho}{2}\Big)\|\hat{x}(\bar{y}) - x_0\|^2 + \frac{1}{\eta_y T}\|\hat{y}(\bar{x}) - y_0\|^2$$

Taking expectation over both sides and recalling that $\mathrm{E}[\|\partial_x f(x, y; \xi)\|^2] \leq M_1^2$ and $\mathrm{E}[\|\partial_y f(x, y; \xi)\|^2] \leq M_2^2$, we have

$$\mathrm{E}[\widehat{\mathrm{Gap}}(\bar{x}, \bar{y})] = \mathrm{E}[\hat{f}(\bar{x}, \hat{y}(\bar{x})) - \hat{f}(\hat{x}(\bar{y}), \bar{y})] \leq \frac{1}{T} \sum_{t=0}^{T-1} \mathrm{E}[\hat{f}(x_t, \hat{y}(\bar{x})) - \hat{f}(\hat{x}(\bar{y}), y_t)]$$

$$\leq \frac{5\eta_x M_1^2}{2} + \frac{5\eta_y M_2^2}{2} + \frac{1}{T}\Big(\frac{1}{\eta_x} + \frac{\rho}{2}\Big)\mathrm{E}[\|\hat{x}(\bar{y}) - x_0\|^2] + \frac{1}{\eta_y T}\mathrm{E}[\|\hat{y}(\bar{x}) - y_0\|^2].$$

$\square$

# H   Proof of Lemma 5

Before proving Lemma 5, we first state the following lemma, whose proof is in the next section.

**Lemma 8.** $\widehat{\mathrm{Gap}}_k(\bar{x}_k, \bar{y}_k)$ *could be lower bounded by the following inequalities*

1) $\widehat{\mathrm{Gap}}_k(\bar{x}_k, \bar{y}_k) \geq (1 - \frac{\gamma}{\rho}(\frac{1}{\alpha} - 1))\widehat{\mathrm{Gap}}_{k+1}(x_0^{k+1}, y_0^{k+1}) + \frac{\gamma}{2}(1 - \frac{1}{1-\alpha})\|x_0^{k+1} - x_0^k\|^2,$

2) $\widehat{\mathrm{Gap}}_k(\bar{x}_k, \bar{y}_k) \geq P(x_0^{k+1}) - P(x_0^k) + \frac{\gamma}{2}\|\bar{x}_k - x_0^k\|^2,$ *where* $P(x) = \max_{y \in Y} f(x, y),$

3) $\widehat{\mathrm{Gap}}_k(\bar{x}_k, \bar{y}_k) \geq \frac{\rho(1-\beta)}{2(\frac{1}{\beta} - 1)}\|x_0^k - \hat{x}_k^*\|^2 - \frac{\rho}{2(\frac{1}{\beta} - 1)}\|\bar{x}_k - x_0^k\|^2,$   (37)

*where* $0 < \alpha \leq 1$ *and* $0 < \beta \leq 1$.

*Proof.* (of Lemma 5)

$$\widehat{\text{Gap}}_k(\bar{x}_k, \bar{y}_k)$$

$$=\frac{1}{10}\widehat{\text{Gap}}_k(\bar{x}_k, \bar{y}_k) + \frac{4}{5}\widehat{\text{Gap}}_k(\bar{x}_k, \bar{y}_k) + \frac{1}{10}\widehat{\text{Gap}}_k(\bar{x}_k, \bar{y}_k)$$

$$\overset{(a)}{\geq} \frac{1}{10}\Big\{\Big(1 - \frac{\gamma}{\rho}(\frac{1}{\alpha} - 1)\Big)\widehat{\text{Gap}}_{k+1}(x_0^{k+1}, y_0^{k+1}) + \frac{\gamma}{2}(1 - \frac{1}{1-\alpha})\|x_0^{k+1} - x_0^k\|^2)\Big\}$$

$$+ \frac{4}{5}\Big\{P(x_0^{k+1}) + \frac{\gamma}{2}\|\bar{x}_k - x_0^k\|^2 - P(x_0^k)\Big\}$$

$$+ \frac{1}{10}\Big\{\frac{\rho(1-\beta)}{2(\frac{1}{\beta} - 1)}\|x_0^k - \hat{x}_k^*\|^2 - \frac{\rho}{2(\frac{1}{\beta} - 1)}\|\bar{x}_k - x_0^k\|^2\Big\}$$

$$= \frac{1}{10}\Big(1 - \frac{\gamma}{\rho}(\frac{1}{\alpha} - 1)\Big)\widehat{\text{Gap}}_{k+1}(x_0^{k+1}, y_0^{k+1}) + \frac{4}{5}(P(x_0^{k+1}) - P(x_0^k))$$

$$+ \Big(\frac{1}{10}\cdot\frac{\gamma}{2}(1 - \frac{1}{1-\alpha}) + \frac{4}{5}\cdot\frac{\gamma}{2} - \frac{1}{10}\cdot\frac{\rho}{2(\frac{1}{\beta} - 1)}\Big)\|\bar{x}_k - x_0^k\|^2$$

$$+ \frac{1}{10}(\frac{\rho(1-\beta)}{2(\frac{1}{\beta} - 1)})\|x_0^k - \hat{x}_k^*\|^2]$$

$$\overset{(b)}{=} \frac{1}{10}(1 - 2(\frac{1}{\frac{5}{6}} - 1))\widehat{\text{Gap}}_{k+1}(x_0^{k+1}, y_0^{k+1}) + \frac{4}{5}(P(x_0^{k+1}) - P(x_0^k))$$

$$+ \Big(\frac{1}{10}\cdot\frac{\gamma}{2}(1 - \frac{1}{1-\frac{5}{6}}) + \frac{4}{5}\cdot\frac{\gamma}{2} - \frac{1}{10}\cdot\frac{\gamma}{4(\frac{1}{2} - 1)}\Big)\|\bar{x}_k - x_0^k\|^2$$

$$+ \frac{1}{10}(\frac{\rho(1-\frac{1}{2})}{2(\frac{1}{\frac{1}{2}} - 1)})\|x_0^k - \hat{x}_k^*\|^2$$

$$= \frac{3}{50}\widehat{\text{Gap}}_{k+1}(x_0^{k+1}, y_0^{k+1})] + \frac{4}{5}(P(x_0^{k+1}) - P(x_0^k))$$

$$+ \frac{\gamma}{8}\|\bar{x}_k - x_0^k\|^2 + \frac{\gamma}{80}\|x_0^k - \hat{x}_k^*\|^2$$

$$\overset{(c)}{\geq} \frac{3}{50}\widehat{\text{Gap}}_{k+1}(x_0^{k+1}, y_0^{k+1}) + \frac{4}{5}(P(x_0^{k+1}) - P(x_0^k))$$

$$+ \frac{\gamma}{80}\|x_0^k - \hat{x}_k^*\|^2, \tag{38}$$

where inequality $(a)$ is due to Lemma 8, inequality $(b)$ is due to the setting of $\gamma = 2\rho$, $\alpha = \frac{5}{6}$, $\beta = \frac{1}{2}$. Inequality $(c)$ is due to $\|\bar{x}_k - x_0^k\|^2 \geq 0$. $\qquad\square$

# I Proof of Lemma 8

*Proof.* Before we prove the three results, we first state two results of Young's inequality as follows

$$\|x - y\|^2 = \|x - z + z - y\|^2 = \|x - z\|^2 + \|z - y\|^2 - 2\langle x - z, y - z\rangle$$

$$\geq \|x - z\|^2 + \|z - y\|^2 - \alpha\|x - z\|^2 - \frac{1}{\alpha}\|y - z\|^2$$

$$= (1 - \alpha)\|x - z\|^2 + (1 - \frac{1}{\alpha})\|z - y\|^2$$

$$\Rightarrow \|x - z\|^2 \leq \frac{1}{1-\alpha}\|x - y\|^2 + \frac{1}{\alpha}\|y - z\|^2 \tag{39}$$

$$\Rightarrow -\|y - z\|^2 \leq -\alpha\|x - z\|^2 + \frac{\alpha}{1-\alpha}\|x - y\|^2 \tag{40}$$

$$\Rightarrow \|x - y\|^2 \geq (1 - \alpha)\|x - z\|^2 + (1 - \frac{1}{\alpha})\|y - z\|^2, \tag{41}$$

where $0 < \alpha \leq 1$.

We first consider the result 1).

$$\widehat{\text{Gap}}_{k+1}(x_0^{k+1}, y_0^{k+1})$$

$$=\hat{f}_{k+1}(x_0^{k+1}, \hat{y}_{k+1}(x_0^{k+1})) - \hat{f}_{k+1}(\hat{x}_{k+1}(y_0^{k+1}), y_0^{k+1})$$

$$=f(x_0^{k+1}, \hat{y}_{k+1}(x_0^{k+1})) + \frac{\gamma}{2}\|x_0^{k+1} - x_0^{k+1}\|^2$$

$$\quad - f(\hat{x}_{k+1}(y_0^{k+1}), y_0^{k+1}) - \frac{\gamma}{2}\|\hat{x}_{k+1}(y_0^{k+1}) - x_0^{k+1}\|^2$$

$$=f(x_0^{k+1}, \hat{y}_{k+1}(x_0^{k+1})) + \frac{\gamma}{2}\|x_0^{k+1} - x_0^k\|^2 - f(\hat{x}_{k+1}(y_0^{k+1}), y_0^{k+1}) - \frac{\gamma}{2}\|\hat{x}_{k+1}(y_0^{k+1}) - x_0^k\|^2$$

$$\quad + \frac{\gamma}{2}\|\hat{x}_{k+1}(y_0^{k+1}) - x_0^k\|^2 - \frac{\gamma}{2}\|\hat{x}_{k+1}(y_0^{k+1}) - x_0^{k+1}\|^2 - \frac{\gamma}{2}\|x_0^{k+1} - x_0^k\|^2$$

$$\overset{(a)}{\leq} f(\bar{x}_k, \hat{y}_{k+1}(\bar{x}_k)) + \frac{\gamma}{2}\|\bar{x}_k - x_0^k\|^2 - f(\hat{x}_{k+1}(\bar{y}_k), \bar{y}_k) - \frac{\gamma}{2}\|\hat{x}_{k+1}(\bar{y}_k) - x_0^k\|^2$$

$$\quad + \frac{\gamma}{2}\left\{\frac{1}{\alpha}\|\hat{x}_{k+1}(y_0^{k+1}) - x_0^{k+1}\|^2 + \frac{1}{1-\alpha}\|x_0^{k+1} - x_0^k\|^2\right\}$$

$$\quad - \frac{\gamma}{2}\|\hat{x}_{k+1}(y_0^{k+1}) - x_0^{k+1}\|^2 - \frac{\gamma}{2}\|x_0^{k+1} - x_0^k\|^2$$

$$=\hat{f}_k(\bar{x}_k, \hat{y}_k(\bar{x}_k)) - \hat{f}_k(\hat{x}_{k+1}(\bar{y}_k), \bar{y}_k)$$

$$\quad + \frac{\gamma}{2}(\frac{1}{\alpha} - 1)\|\hat{x}_{k+1}(y_0^{k+1}) - x_0^{k+1}\|^2 + \frac{\gamma}{2}(\frac{1}{1-\alpha} - 1)\|x_0^{k+1} - x_0^k\|^2$$

$$\overset{(b)}{\leq} \hat{f}_k(\bar{x}_k, \hat{y}_k(\bar{x}_k)) - \hat{f}_k(\hat{x}_{k+1}(\bar{y}_k), \bar{y}_k)$$

$$\quad + \frac{\gamma}{2}(\frac{1}{\alpha} - 1)\frac{2}{\rho}(\hat{f}_{k+1}(x_0^{k+1}, y_0^{k+1}) - \hat{f}_{k+1}(\hat{x}_{k+1}(y_0^{k+1}), y_0^{k+1})) + \frac{\gamma}{2}(\frac{1}{1-\alpha} - 1)\|x_0^{k+1} - x_0^k\|^2$$

$$\overset{(c)}{\leq} \widehat{\text{Gap}}_k(\bar{x}_k, \bar{y}_k) + \frac{\gamma}{\rho}(\frac{1}{\alpha} - 1)\widehat{\text{Gap}}_{k+1}(x_0^{k+1}, y_0^{k+1}) + \frac{\gamma}{2}(\frac{1}{1-\alpha} - 1)\|x_0^{k+1} - x_0^k\|^2,$$

where inequality $(a)$ is due to (39) $(0 < \alpha \leq 1)$. Inequality $(b)$ is due to $\rho$-strong convexity of $\hat{f}_{k+1}(x, y_0^{k+1})$ in $x$ and optimality at $\hat{x}_{k+1}(y_0^{k+1})$. Inequality $(c)$ is due to $\hat{f}_k(\hat{x}_{k+1}(\bar{y}_k), \bar{y}_k) \geq \hat{f}_k(\hat{x}_k(\bar{y}_k), \bar{y}_k)$ and $\hat{f}_{k+1}(x_0^{k+1}, y_0^{k+1}) \leq \hat{f}_{k+1}(x_0^{k+1}, \hat{y}_{k+1}(x_0^{k+1}))$.

Re-organizing the above inequality, we have

$$\widehat{\text{Gap}}_k(\bar{x}_k, \bar{y}_k) \geq (1 - \frac{\gamma}{\rho}(\frac{1}{\alpha} - 1))\widehat{\text{Gap}}_{k+1}(x_0^{k+1}, y_0^{k+1}) + \frac{\gamma}{2}(1 - \frac{1}{1-\alpha})\|x_0^{k+1} - x_0^k\|^2,$$

which proves result 1).

Then we turn to result 2) as follows.

$$\widehat{\text{Gap}}_k(\bar{x}_k, \bar{y}_k) = \hat{f}_k(\bar{x}_k, \hat{y}_k(\bar{x}_k)) - \hat{f}_k(\hat{x}_k(\bar{y}_k), \bar{y}_k)$$

$$\geq \hat{f}_k(\bar{x}_k, \hat{y}_k(\bar{x}_k)) - \hat{f}_k(x_0^k, \bar{y}_k)$$

$$\geq \hat{f}_k(\bar{x}_k, \hat{y}_k(\bar{x}_k)) - \hat{f}_k(x_0^k, \hat{y}_k(x_0^k))$$

$$= f(\bar{x}_k, \hat{y}_k(\bar{x}_k)) + \frac{\gamma}{2}\|\bar{x}_k - x_0^k\|^2 - f(x_0^k, \hat{y}_k(x_0^k)) - 0$$

$$= f(x_0^{k+1}, \hat{y}_k(x_0^{k+1})) + \frac{\gamma}{2}\|\bar{x}_k - x_0^k\|^2 - f(x_0^k, \hat{y}_k(x_0^k))$$

$$= P(x_0^{k+1}) - P(x_0^k) + \frac{\gamma}{2}\|\bar{x}_k - x_0^k\|^2,$$

which proves result 2).

Result 3) can be proved as follows

$$\|\bar{x}_k - x_0^k\|^2 \overset{(a)}{\geq} (1-\beta)\|x_0^k - \hat{x}_k^*\|^2 + (1-\frac{1}{\beta})\|\hat{x}_k^* - \bar{x}_k\|^2$$

$$\overset{(b)}{\geq} (1-\beta)\|x_0^k - \hat{x}_k^*\|^2 + (1-\frac{1}{\beta})\frac{2}{\rho}(\hat{f}_k(\bar{x}_k, \hat{y}_k^*) - \hat{f}_k(\hat{x}_k^*, \hat{y}_k^*))$$

$$\overset{(c)}{\geq} (1-\beta)\|x_0^k - \hat{x}_k^*\|^2 + (1-\frac{1}{\beta})\frac{2}{\rho}\widehat{\text{Gap}}_k(\bar{x}_k, \bar{y}_k))$$

$$\Rightarrow \widehat{\text{Gap}}_k(\bar{x}_k, \bar{y}_k) \geq \frac{\rho(1-\beta)}{2(\frac{1}{\beta}-1)}\|x_0^k - \hat{x}_k^*\|^2 - \frac{\rho}{2(\frac{1}{\beta}-1)}\|\bar{x}_k - x_0^k\|^2,$$

where inequality $(a)$ is due to (41) and $0 < \beta \leq 1$. Inequality $(b)$ is due to $\rho$-storng convexity of $\hat{f}_k$ in $x$. Inequality $(c)$ is due to $0 < \beta \leq 1$ and

$$\hat{f}_k(\bar{x}_k, \hat{y}_k^*) - \hat{f}_k(\hat{x}_k^*, \hat{y}_k^*) \leq \hat{f}_k(\bar{x}_k, \hat{y}_k(\bar{x}_k)) - \hat{f}_k(\hat{x}_k^*, \bar{y}_k)$$

$$\leq \hat{f}_k(\bar{x}_k, \hat{y}_k(\bar{x}_k)) - \hat{f}_k(\hat{x}_k(\bar{y}_k), \bar{y}_k)$$

$$= \widehat{\text{Gap}}_k(\bar{x}_k, \bar{y}_k).$$

$\square$