[Reviews · NeurIPS 2020]

Review 1

Summary and Contributions: This paper proposes an epoch-wise stochastic gradient descent ascent method ( Epoch-GDA) for solving strongly convex strongly concave (SCSC) min-max problems. Moreover, it studies the convergence properties of the Epoch-GDA method, which achieve an optimal rate of $O(1/T)$ for the duality gap of general SCSC min-max problems without smoothness and funcation’s structure assumptions.

Strengths: This paper provides a sharp convergence analysis of the Epoch-GDA method for solving strongly convex strongly concave (SCSC) min-max problems without smoothness and funcation’s structure assumptions. At the same time, it also provides a sharp convergence analysis of the Epoch-GDA method for solving weakly-convex strongly-concave (WCSC) min-max problems.

Weaknesses: The paper should provide some experimental results to demonstrate the efficiency of the proposed algorithms.

Correctness: All methods in the paper are correct.

Clarity: This paper is well written.

Relation to Prior Work: The related work part of this paper clearly discuss differences between the proposed methods and the previous methods.

Reproducibility: Yes

Additional Feedback: Some comments are given as follows: C1: Epoch GDA seems very similar to Epoch GD in terms of algorithm presentation and analysis statement. Can the authors highlight the key difference between those two algorithms? C2: When comparing with [36] and [32], the authors claim deterministic updates are present in both algorithms. Why does this paper get around of the deterministic updates? C3: When stating the lower bound in Remark 2, O(1/T) should be \Omega(1/T). %%%%%%%%%%%%%%%%%%%%%% Thanks for your responses. I still maintain earlier positive review and recommend acceptance.


Review 2

Summary and Contributions: This paper extends the work of [13], who provide an algorithm (Epoch-GD) which returns a $O(1/T)$-approximate solution (referred to here as the objective gap) for minimization of (possibly non-smooth) strongly convex functions after $T$ stochastic gradient oracle calls. The main result in the current paper extends this result to a $O(1/T)$ convergence bound for the duality gap of a (possibly non-smooth) strongly convex-strongly concave min-max problem (instead of the objective gap), for a min-max version of this algorithm (Epoch-GDA). They also provide a related bound for finding approximate stationary points in weakly convex-strongly concave problems.

Strengths: The main strength of the work is that they extend the $O(1/T)$ bound which was previously obtained for the objective gap, to the duality gap for min-max problems. Moreover, the authors also provide a bound for convergence to an approximate stationary point for weakly convex-strongly concave problems.

Weaknesses: A weakness of the paper is that the key technical contribution (Lemma 1), which connects the duality gap to Euclidean distance, is fairly simple to prove, as a consequence of the strong convexity and strong concavity assumptions. For this reason, it is not clear why (in the strongly convex-strongly concave setting) extending the O(1/T) bound from the objective gap to the duality gap is an important technical contribution.

Correctness: Although I have not checked the proof itself, the overview of the proof provided by the authors seems to be correct.

Clarity: The paper seems to be clearly written.

Relation to Prior Work: The relation to prior work is clearly discussed.

Reproducibility: Yes

Additional Feedback: I am satisfied with the authors' explanations, and have adjusted my score accordingly.


Review 3

Summary and Contributions: %%%%%%%%%%%% Update after rebuttal %%%%%%%%%%%% I would like to thank the authors for their explanation and I will be glad to see the revisions promised by the authors appear in a new version of the paper. ---------------------------------------------------------------------------------------------------- In the paper, the authors propose the first algorithm that achieves the O(1/T) optimal rate for the duality gap (or the Nikaido-Isoda function) in stochastic strongly-convex-strongly-concave saddle-point problems without resorting to either smoothness or specific structure of the problem. The algorithm itself is a straightforward combination of Epoch-SGD for minimization and the well-known gradient-descent-ascent method for minimax problem. At the heart of the analysis is a new lemma that relates the duality gap to some distance measures between two points. Basing on these results the authors also derive new algorithms for stochastic non-smooth weakly-convex-strongly-concave saddle-point problems with improved complexity.

Strengths: Recently, there has been an abundance of emerging work that studies the performance of first-order methods (GDA/EG/OG) in saddle-point problems under different assumptions. I believe these fundamental problems are important for us to build a global picture of how various algorithms designed for saddle-point optimization behave in different settings. Importantly, this paper investigates the relatively less explored non-smooth setup and present simple algorithms with proper analysis, which is a nice contribution to the field.

Weaknesses: Please refer to the following parts for detailed comments.

Correctness: I did not check the proofs in detail, but the methodology look reasonable and several techniques that are commonly used for the analysis of stochastic saddle-point problems are also employed in the paper. I just want to mention that for Theorem 2, I fail to see why we always have dist(0,∂P(\hat{x}_𝜏^*)) <= γ\|\hat{x}_𝜏^*-x_0^𝜏\|. I fully understand that this is true when the minimizer of the regularized problem lies in interior of X. However, considering the case where x lies on the boundary, I suppose we may need to add the normal cone to the ∂P term to so that the statement handles this case as well?

Clarity: The paper is overall well-organized, but that there seems to be too much redundancy in the description of results. In particular, Remark 3 is basically repeating what is mentioned in the paragraphs before Theorem 2, and the authors could consider removing either the remark or the corresponding paragraph to save some space. I also spot several typos. For example, the is a confusion in the use of the words gradient and subgradient (I believe all of them should be subgradient). In lemma 1, x* and y* are defined in the statement but only used in the proof. In line 217, one should write \hat{f}_k instead of f_k, and in line 138 in the definition of the limiting subgradient one should put v ∈ R^d in the place of v_k ∈ R^d. Finally, the authors could also consider providing more intuition behind the definition of a nearly ε-stationary point (for example, by mentioning the Moreau envelope) or pointing the readers to a reference for this concept.

Relation to Prior Work: The paper does a great job in summarizing recent works working on similar topics. Nonetheless, I find some important references are missing. In particular, the convergence in O(1/k) in terms of mean squared distance for non-smooth strongly-monotone variational inequalities (a generalization of strongly-convex-strongly-concave saddle-point problem) was shown in [1]. While the algorithmic idea is very different and the actual contribution also differs (dual gap versus distance to solution as the convergence measure), I think given the similarity of the results, this paper should be mentioned by the authors. [1] Yousefian, F., Nedić, A., & Shanbhag, U. V. (2015). Self-tuned stochastic approximation schemes for non-Lipschitzian stochastic multi-user optimization and Nash games. IEEE Transactions on Automatic Control, 61(7), 1753-1766.

Reproducibility: Yes

Additional Feedback: In Theorem 1 of the paper, a high probability result is provided while in Theorem 2 a bound in expectation is proved. Could you comment on this inconsistency? I am curious to know if there is any fundamental reason behind it.

[Author Response · NeurIPS 2020]

**Reviewer 1:** Q1: This paper should provide experimental results?

A: We believe it is important to deliver the message that our proof is novel that addresses the open problem for strongly-convex-strongly-concave minimization. Hence, we emphasize on theoretical analysis in this paper. On the other hand, previous studies have provided the numerical experiments on the state-of-the-art algorithms that are highly related to our Epoch-GDA, e.g., [36,32]. There is also a following-up work that uses a similar idea and has promising experimental results [ref1]. We will consider adding some experiments in the long version.

[ref1] Guo et al. "Fast Objective and Duality Gap Convergence for Non-convex Strongly-concave Min-max Problems". arXiv 2020.

Q2: Key difference of algorithm and analysis between the proposed Epoch-GDA and Epoch-GD?

A: The update of Epoch-GDA can be seen as a primal-dual variant of Epoch-GD. In terms of analysis, there is key difference between Epoch-GDA and Epoch-GD. In particular, Epoch-GD bounds the primal gap, while Epoch-GDA bounds the duality gap. Note that bounding the duality gap of a min-max problem is fundamentally more difficult than bounding the primal gap of a minimization problem. The difference between our analysis and that of Hazan & Kale is very subtle. Particularly, in Hazan & Kale, they used the fact that the primal gap at a solution $x_{k+1}$ from stage $k+1$ can be bounded by the distance between a solution $x_k$ from stage $k$ and the optimal solution i.e., $\|x_k - x_*\|$, which can be further bounded by the primal gap at $x_k$ using strong convexity. However, in our case, the duality gap at a solution $(x_{k+1}, y_{k+1})$ from stage $k+1$ cannot be bounded by the distance between $(x_k, y_k)$ from stage $k$ and the optimal solution. Instead, they are bounded by the distance from $(x_k, y_k)$ to the corresponding optimal solutions to the minimization and maximization defined at $(x_{k+1}, y_{k+1})$, i.e., $\|x_k - \hat{x}_R(y_{k+1})\|$ and $\|y_k - \hat{y}_R(x_{k+1})\|$ (cf. the key Lemma 3). More importantly, we have to show that this distance is strictly less than the imposed radius $R$ such that adding the bounded ball preserves the duality gap of the original problem. Please note that such interior-point argument is very important and is not necessary in Hazan & Kale.

Q3: Why avoiding deterministic updates as in [36,32]?

A: The reason is two-fold: (i) the deterministic updates in [36, 32] require a specific form of objective function; hence by avoiding deterministic update we are able to handle more general problems without scarifying the complexity; (ii) the deterministic updates in [36, 32] have additional computional overhead, which usually needs to pass all data in machine learning applications. A key difference from [36, 32] is that we use the recursion on the duality gap as for convergence analysis, while [36,32] use the recursion on the primal objective gap for convergence analysis.

**Reviewer 2:** Q1: Key technical contribution (Lemma 1) is simple to prove, so unclear why it is important to extend Epoch-GD for SC min problem to Epoch-GDA for SCSC min-max problem.

A: We agree Lemma 1 is simple to prove. But **the key for proving the fast rate of duality gap for SCSC problems lies at Lemma 3**, which proves that the duality gap of the problem defined with the ball constraint is equal to the original duality gap. This proof is subtly different from that of Epoch-GD. Please refer to response to Q2 of reviewer 1.

**Reviewer 3:** Thanks for pointing out the relevant reference. We will add it in the revision.

Q1: Why we always have dist$(0, \partial P(\hat{x}_\tau^*)) \leq \gamma \|\hat{x}_\tau^* - x_0^\tau\|$ in Theorem 2?

A: Thanks for pointing this out. Indeed, we should use $\hat{P} = P + \mathbb{I}_X$ in place of $P$ in the above inequality, where $\mathbb{I}_X$ is the indicator function of the set $X$, which gives us the desired result. To prove this, let us consider an unconstrained $\rho$-weakly convex function $\psi(x)$ and a reference point $\tilde{x}$, $f(x) = \psi(x) + \frac{\gamma}{2}\|x - \tilde{x}\|^2$ is $(\gamma - \rho)$-strongly convex. Hence, we have the unique optimal solution of $\min_x f(x)$, say $\hat{x}$, then the optimality condition gives that $0 \in \partial\psi(\hat{x}) + \gamma(\hat{x} - \tilde{x})$, i.e., $\gamma(\tilde{x} - \hat{x}) \in \partial\psi(\hat{x})$, which means dist$(0, \partial\psi(\hat{x})) \leq \gamma\|\hat{x} - \tilde{x}\|$. Applying this argument to $P(x) + \mathbb{I}_X$ leads to the corrected inequality.

Q2: Theorem 1 provides a high probability result, while Theorem 2 proves a bound in expectation?

A: Thanks for noticing this difference. We prove the expectation result for WCSC in Theorem 2 for consistency with previous results [32]. Indeed, we followed your suggestion and found that Thm. 2 can be also extended to high-probability result. The key idea is similar to that for proving Thm. 1. In particular, we can prove a high-probability result of Lemma 4 similar to Lemma 2. Then by appropriately setting the radius $R_k$ according to $\eta_k$ and $T_k$ we can able to prove a similar result as in Lemma 3, which leads to a high-probability upper bound for the duality gap of $f_k(x, y)$. From this point, we can prove the high-prob convergence for the WCSC similar to the existing proof of Thm. 2 except replacing expectation result with high-probability result. We will discuss this in the revision.



[Meta-Review · NeurIPS 2020]

The main result in the paper extends a classical result of Hazan et al to a $O(1/T)$ convergence bound for the duality gap of non-smooth strongly convex-strongly concave min-max problem (instead of the objective gap), proposing a min-max adaptation of the algorithm (Epoch-GDA). They also provide a related bound for finding approximate stationary points in weakly convex-strongly concave problems. Overall the reviewers found the contribution to be a significant and challenging extension over the existing result of Hazan et al. with specific challenges to be overcome in the duality gap version. The authors are strongly recommended to make the promised revisions in the rebutttal as they will embellish the paper.